# Subgroup Robustness Grows On Trees: An Empirical Baseline Investigation

**Josh Gardner**[1]         **Zoran Popović**[1]         **Ludwig Schmidt**[1,2]

[1] **University of Washington**     [2] **Allen Institute for AI**
{jpgard, zoran, schmidt}@cs.washington.edu

## Abstract

Researchers have proposed many methods for fair and robust machine learning, but comprehensive empirical evaluation of their subgroup robustness is lacking. In this work, we address this gap in the context of tabular data, where sensitive subgroups are clearly-defined, real-world fairness problems abound, and prior works often do not compare to state-of-the-art tree-based models as baselines. We conduct an empirical comparison of several previously-proposed methods for fair and robust learning alongside state-of-the-art tree-based methods and other baselines. Via experiments with more than 340,000 model configurations on eight datasets, we show that tree-based methods have strong subgroup robustness, even when compared to robustness- and fairness-enhancing methods. Moreover, the best tree-based models tend to show good performance over a range of metrics, while robust or group-fair models can show brittleness, with significant performance differences across different metrics for a fixed model. We also demonstrate that tree-based models show less sensitivity to hyperparameter configurations, and are less costly to train. Our work suggests that tree-based ensemble models make an effective baseline for tabular data, and are a sensible default when subgroup robustness is desired.[1]

## 1   Introduction

Over the past decade, the field of machine learning (ML) has seen a dramatic expansion along two related lines. On one hand, concerns about the fairness of ML models, and more broadly their performance on data outside the training distribution, have grown [50]. Both theoretical and empirical works have raised these concerns, demonstrating the vulnerability of models to learn biases from data or suffer performance drops under distribution shifts [44, 66, 35]. On the other hand, an abundance of methods have been proposed to address these limitations. These include *fairness* methods used to equalize metrics across groups, as well as *distributional robustness* methods which optimize for a worst-case distribution within a bounded distance of the training distribution.

Our work begins from the observation that, while "fairness" and "robustness" are distinct concepts, methods in both areas often have similar goals. In particular, they share two (sometimes implicit) goals: First, models should have low *disparity* (variation in performance over all subgroups should be minimized; cf. [75, 17, 25, 43]). Second, models should have good *worst-group performance* (the lowest accuracy over all subgroups should be maximized; cf. [74, 62]). As a consequence of these two criteria, both fairness and robustness typically entail improving *subgroup robustness*.

---

[1]See https://github.com/jpgard/subgroup-robustness-grows-on-trees for code to reproduce our experiments and detailed experimental results.

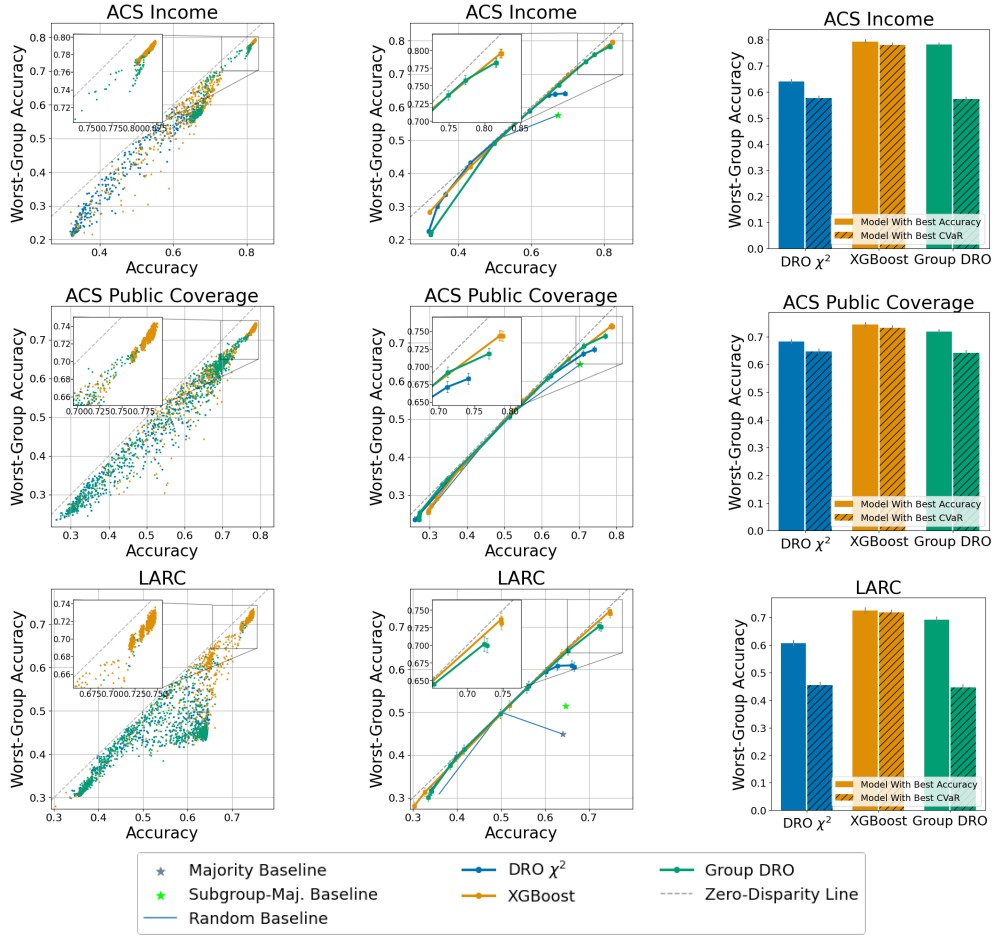

Figure 1: Results from three datasets in our study. (a) Left: Tree-based methods such as XG-Boost show similar subgroup robustness, with sometimes better overall performance, as robustness-enhancing or disparity-mitigation methods. (b) Center: Model performance frontiers corresponding to (a) show similar accuracy-robustness frontiers for XGBoost and DRO/Group DRO. (c) Right: Tree-based methods' worst-group accuracy is robust to model selection metrics.

Our work focuses on subgroup robustness in the context of *tabular* data, for several reasons. First, tabular data is widely used in practice, being the most common format in areas with important fairness impacts such as medicine, finance, and recommender systems [38, 64]. Second, tabular data often directly encodes sensitive attributes with respect to which subgroup robustness is desired (race, gender, income level), and these attributes are often declared by the individuals represented in the dataset. This is an important difference compared to crowdsourced group annotations (e.g., gender or race for image datasets), which can be unreliable [21] or fail to reflect the lived experiences of the individuals represented [19]. Finally, tabular data is challenging to model. It is often heterogeneous, containing mixed data types; it lacks the spatial, linguistic, or temporal structures common to other data (image, text, audio) where machine learning has seen dramatic progress over the past decade; and it can be relatively small, containing personal information which cannot be scraped at Internet scale.

In this work, we conduct a series of experiments to jointly compare methods from the fairness and robustness literature. Despite the fact that many works often use similar metrics and datasets, there is a lack of large-scale empirical studies comparing relevant methods to each other, and to strong baselines. This lack of effective baselines is particularly concerning in light of recent works across several other areas of machine learning which have shown that simple baselines can perform surprisingly well against state-of-the-art techniques [47].

Our results show that modern tabular-data baselines can outperform even well-tuned state-of-the-art robustness- or fairness-enhancing methods with a fraction of the computational cost. Our analysis includes a set of tree-based models which achieve strong performance on tabular data but are absent from prior works on robustness, which largely only compare to deep learning-based approaches despite their tendency to perform poorly on tabular data [38]. In our experiments, we endeavor to tune both the "fair" or "robust" models and a representative suite of baseline methods for tabular data to obtain estimates of the empirical accuracy-subgroup robustness Pareto frontier for each model. To this end, we explore a wide space of architectures, regularization schemes, and training hyperparameters, across eight tabular datasets and 17 model classes. Our work is an empirical contribution concerned not with proposing new methods, but with conducting a novel and rigorous evaluation of existing methods for learning subgroup-robust models from tabular data. Our contributions are as follows:

**Trees are subgroup-robust learners:** While previous works have shown that modern tree-based ensembles have strong average-group performance (see Section 2.3), we show that these models are also surprisingly robust to subgroup shift. Tree-based models achieve this robustness despite lacking any explicit robustness intervention and using only average-case classification losses, achieving competitive or better performance against state-of-the-art methods for fairness and robustness (Figure 1a,1b) over a large hyperparameter and model search space. To support the above analysis, we apply several techniques for the analysis of hyperparameter sweeps such as model performance frontier curves (Figure 1b,3) and hyperparameter sensitivity analysis (Figures 7, 8, 9).

**Trees show show consistent performance across accuracy and robustness metrics:** We empirically investigate the relationship between three types of metrics (accuracy, robustness, fairness). We show that metrics within these groups often agree, but the relationships across metric groups (i.e. robust CVaR risk and accuracy) are inconsistent. One consequence of this finding is that model selection techniques affect tree-based models and "robust" or fairness-enhancing methods differently: tree-based methods show consistent subgroup robustness across metrics (Figure 1c), while robust and fairness-enhancing methods are "metric-brittle" and tend to display poor subgroup robustness according to other metrics (e.g. worst-group accuracy).

**Trees are less sensitive to hyperparameters and less costly to train:** We show that, in addition to their improved robustness when fully tuned, trees also require less tuning and are less costly to train. We demonstrate that trees have decreased sensitivity to hyperparameters – even when accounting for hyperparameter settings which were part of our initial grid but performed poorly across all datasets – and that these models require considerably less compute and financial cost to train when compared to deep learning-based robust learning techniques.

In particular, our results highlight the importance of the underlying model class for subgroup robustness in tabular data. Our results suggest that subgroup robustness interventions based on the loss-based perspective alone may be limited, particularly because existing loss-based methods for robustness are incompatible with nondifferentiable functions such as tree-based ensembles. We provide further discussion in Section 7.

We provide code to reproduce our experiments, along with an interactive tool to explore the best-performing hyperparameter configurations, at `https://github.com/jpgard/subgroup-robustness-grows-on-trees`.

## 2 Related Work

### 2.1 Fairness-Enhancing Methods for Supervised Learning

A wide variety of works have addressed fairness in the context of machine learning, where "fairness" is often measured by the equalization of some metric over groups [5, 18]. Most methods can be characterized as performing either pre- in-, or post-processing, which attempt to ensure fairness considerations are met in different stages of the modeling lifecycle. *Preprocessing* [73, 15] attempts to modify the input data to meet fairness criteria, while preserving the structure of the inputs and their relationship to the prediction target. *Inprocessing* uses a modified training procedure to explicitly optimize for a fairness-aware objective during model training. This includes using constrained or reduction-based optimization [1, 72], or including explicit regularizers designed to encourage fairness [7, 11, 43, 56]. *Postprocessing* [32, 53] operates on the predictions of a model, modifying them to achieve fairness criteria.

The impact of fairness under subpopulation shift is analyzed in [48], which shows theoretically that enforcing fairness during training can harm or improve the model, under certain conditions, and [65] conducts a causal analysis of fairness under covariate shift. Perhaps the work most closely related to the current study is Friedler et al. [26], which evaluates a set of pre-, in-, and postprocessing algorithms, demonstrating that many of these techniques show instability (variations in performance over different train-test splits) and that several fairness metrics are empirically correlated. Friedler et al. [26] do not, however, evaluate robust learning techniques, modern tree-based techniques (besides CART), or neural methods. Empirical evaluation of recent methods is needed.

## 2.2   Distributionally Robust Learning

While efforts in robust optimization date back several decades [10], recent works have adapted and extended robust learning approaches to deep learning. For example, several variants of distributionally robust optimization (DRO) have been proposed for the training of neural networks with robustness guarantees [36, 46, 62, 61, 74, 25]. While these approaches are not always explicitly oriented toward fairness, they are frequently evaluated in terms of their performance benefits for minimizing performance gaps between sensitive subgroups in real-world data [74, 62, 33].

A particular form of shift (and robustness) evaluated by these works is *subgroup shift* (also called subpopulation shift) [50], where the balance of subgroups in the data shifts between training and testing. Evaluating performance on individual subgroups is an extreme version of subgroup shift. In the context of fairness, these subgroups are often defined by one or more discrete sensitive attributes, such as race, gender, or income level, and intersections between these sensitive groups can often identify groups most susceptible to performance disparities [14].

## 2.3   Models for Tabular Data

Deep learning-based approaches largely have not achieved the transformative performance gains on tabular data that they have with other data modalities such as images, text, and audio [38, 13, 64]. The existing state-of-the-art for learning with tabular data is widely acknowledged to be tree-based methods, and in particular gradient-boosted trees [31, 54, 64, 13]. This includes GBM [27, 28], LightGBM [40], XGBoost [16], and CatBoost [22, 55], which often show only small differences in performance between them. Several deep learning-based approaches have recently attempted to close the gap of deep learning-based solutions for tabular data [38, 2, 37, 54, 39]. However, empirical analyses have shown that the reported performance of these methods does not generalize well to other datasets, and gradient boosting methods still achieve better performance with less tuning and a considerably smaller computational budget. For example, [64] and [13] both show, in separate analyses, that gradient boosting consistently outperforms these state-of-the-art tabular deep learning methods across several datasets, and tree-based methods achieve competitive performance with the deep learning approaches proposed in [29, 37].

The overall predictive performance of these tabular methods is evaluated in e.g. [13, 29, 38, 54, 64]. However, the existing literature does not investigate the fairness properties, subgroup performance, or robustness to subgroup shift of tabular data models; nor does it compare to subgroup-robust learning methods. Additionally, despite the widely-known strong performance of these models on tabular data, we are aware of no prior work which compares tree-based methods to robust neural network-based learners, despite the fact that the latter are commonly evaluated on tabular data (e.g. [36, 43, 74, 62]). Notably, [1] evaluates GBM in conjunction with their proposed inprocessing method, but does not evaluate subpopulation robustness; [20] evaluates GBM with fairness methods but does not compare to robust methods and does not tune the GBM with fairness methods.

## 3   Setup

Our work is primarily concerned with empirically evaluating the sensitivity of various supervised learning methods to subpopulation shift. Below, we introduce our formal model, and then describe the main axes of variation in our experiments. This includes $(i)$ a large set of models, many of which have not been directly compared in previous works; $(ii)$ the hyperparameters used for each model; $(iii)$ a suite of eight real-world fairness datasets; and $(iv)$ a set of evaluation metrics used across the disparate literature on robustness, subpopulation shift, and fairness in machine learning.

## 3.1 Preliminaries

Our work evaluates the task of learning a model $f_\theta \in \mathcal{F}$, a function from model class $\mathcal{F}$ parameterized by $\theta$. The parameters are learned from a dataset $D := (x_i, y_i)_{i=1}^n \sim \mathcal{P}$ where $\mathcal{P}$ is the data-generating distribution. This matches the case, most common in practice, where a single model is learned by estimating $min_\theta \mathbb{E}(\mathcal{L}(y, f_\theta(x)))$ and deployed for all users. We evaluate the binary classification context, where $y \in \{0, 1\}$ and $f_\theta(x) = \hat{y} \in [0, 1]$ is the score assigned by $f$ to the outcome $y_i = 1$. Each $x_i \in \mathbb{R}^d$ can be partitioned into $x_i = (x_{i,1}, \ldots, x_{i,d-1}, a)$ where $a$ is a sensitive attribute. For notational convenience, we represent $a$ as a single feature here, but in our data, $a$ is typically defined as the concatenation of multiple binary sensitive attributes (e.g. gender = female, race = white). Let $D_a := \{(x_i, y_i) \sim \mathcal{P} | a_i = a\}$ denote the subgroup of the dataset with a given sensitive identity.

Following many of the previous works in both fairness [18] and robustness [74, 43], our analysis focuses on the performance of $f_\theta$ on each $D_a$ for all $a \in \mathcal{A}$. We are particularly interested in the worst-group performance and the loss disparity, respectively defined as

$$\mathcal{L}_{\text{WorstGroup}} := \max_{a \in \mathcal{A}} \mathbb{E}_{D \sim \mathcal{P}}\big[\mathcal{L}_a(f_\theta)\big] \quad \text{and} \quad \mathcal{L}_{\text{DISP}} := \max_{a, a' \in \mathcal{A}} \mathbb{E}_{D \sim \mathcal{P}}\big[|\mathcal{L}_a(f_\theta) - \mathcal{L}_{a'}(f_\theta)|\big] \quad (1)$$

. These metrics can be thought of as assessing the sensitivity of $f_\theta$ to subgroup shift, evaluating the worst-group shift from $D$ to $D_a$ (note that subgroup shift is itself a form of covariate shift [50]).

## 3.2 Models

Our goal is to conduct a thorough empirical comparison of the subgroup robustness of a suite of methods for tabular data. We provide a more detailed description of all models used in Section B.

- **Fairness-enhancing models**: We evaluate the LFR preprocessing method of [73]; the inprocessing method of[1], and the postprocessing method of [32]. As in [1, 20], we use GBM as the base learner for each model; however, unlike [20], we also tune the base learner parameters.

- **Robust models:** We evaluate both the CVaR and $\chi^2$ constraint forms of DRO via [46]; the CVaR and $\chi^2$ constraint forms of DORO [74]; Group DRO [62]; Marginal DRO [24]; and Maximum Weighted Loss Discrepancy [43]. Each robust optimization method is used with a multilayer perceptron (MLP), as in most previous works when using tabular data, e.g. [43, 46, 62, 74]. We note that the use of these losses requires performing optimization over a continuous function (such as a neural network), which makes these loss-based robustness interventions incompatible with existing training procedures for e.g. tree-based models.

- **Tree-based models:** We evaluate GBM, LightGBM [40], XGBoost [16], and Random Forest models.

- **Baseline models:** As baselines, we include $L_2$-regularized logistic regression, Support Vector Machines, and fully-connected neural networks (MLP) with standard (non-DRO) ERM optimization.

## 3.3 Hyperparameter Sweeps

For each model, we conduct a grid search over a large set of hyperparameters. We give the complete set of hyperparameters tuned for each model in Section F.

Our hyperparameter search for each model is extensive by design, in order to ensure a reliable comparison of the best-performing models from each class. We expand the initial grid for continuous hyperparameters when a model on the Pareto frontier is at the edge of the grid. When one method includes another as a "base" learner (e.g. LFR preprocessing with GBM, or DRO with MLP), we include the full tuning grid for the base model (i.e. we explore the cross-product of all MLP hyperparameters with all DRO hyperparameters).

We note that previous works tend to either compare against a fixed baseline architecture and training hyperparameters (e.g. [74, 38]), perform manual tuning ([46]), or do not tune hyperparameters of the base model when using fairness-enhancing techniques ([20]).

## 3.4 Metrics

One goal of our work is to assess the empirical relationship between the model evaluation *metrics* used in the robustness, fairness, and classification literatures. Differences in the *training* objectives used across the various fair and robust models in prior work make such a comparison particularly useful. The relationship between these diverse evaluation metrics is not explored in existing work, where several different metrics have been used to compare the performance of models, evaluate their fairness with respect to subgroups, and measure their robustness to various shifts or outliers. We draw several metrics from prior works and report them for our experiments. We also explore the empirical relationships between different metrics.

For each metric $\mathcal{L}$ (e.g. loss, accuracy, Equalized Odds), we can not only measure the overall empirical performance of a model, but we can measure the *worst-group* loss and *disparity* over subgroups of the data, as defined in Equation (1). Worst-group and disparity measures, for various formulations of the loss functions above, are a widely-used measure of fairness and robustness [12, 74, 62, 61, 33, 44]. In particular, we use the $\mathcal{L}_{\text{DISP}}$ and $\mathcal{L}_{\text{WorstGroup}}$ with accuracy as the loss function. Accuracy is widely used in practice, directly interpretable, and invariant to rescaling of the model's predictions.

We also use the **CVaR risk** metric from the robustness literature. CVaR risk is measured over a set of inputs $D = (x_i, y_i)_{i=1}^N \sim \mathcal{P}$ and measures the worst-case weighted loss, according to some loss function $\ell$, at level $\alpha$ over the inputs in $D$:

$$\mathcal{L}_{\text{CVaR}}(D, \mathcal{P}) \coloneqq \sup_{q \in \Delta^N} \sum_{i=1}^N q_i \ell(f_\theta; x) \quad \text{s.t. } ||q||_\infty \leq \frac{1}{\alpha N} \tag{2}$$

(2) is the risk function optimized by the DRO CVaR model [46], but it is also used more widely as a measure of the tail risk of a classifier (cf. [74]).

Additional fairness and robustness metrics are reported in Section G and defined in Section C.

| Dataset | $n$ | $d$ | Target | Sensitive Attributes |
|---|---|---|---|---|
| ACS Income | 499,350 | 20 | Income $\geq$ 56k | Race, Sex |
| ACS Public Coverage | 341,487 | 19 | Public Ins. Coverage | Race, Sex |
| Adult | 48,845 | 14 | Income $\geq$ 56k | Race, Sex |
| Behavioral Risk Factors Surveillance System (BRFSS) | 175,745 | 28 | Diabetes | Race, Sex |
| Communities & Crime | 1994 | 113 | High Crime | Income Level, Race |
| COMPAS | 7,215 | 10 | Recidivism | Race, Sex |
| German Credit | 1,000 | 22 | Credit Risk | Age, Sex |
| Learning Analytics Architecture (LARC) | 169,032 | 26 | At-Risk (Grade) | URM Status, Sex |

Table 1: Overview of datasets used.

## 3.5 Datasets

We evaluate the 17 models over eight datasets covering a variety of prediction tasks and domains. We use two binary sensitive attributes from each dataset, for a total of four nonoverlapping subgroups in each dataset. A summary of the datasets used in this work is given in Table 1. We provide additional details on each dataset, along with critical framing and context regarding these prediction tasks and their representations of individuals, in Section A.

# 4 Results: Tree Models are Subgroup-Robust Learners

Our main finding is that tree-based models are subgroup-robust learners. Tree-based models match or exceed the performance of distributionally robust and fairness-enhancing models in terms of best overall accuracy, worst-group accuracy, and accuracy disparity on all datasets evaluated.

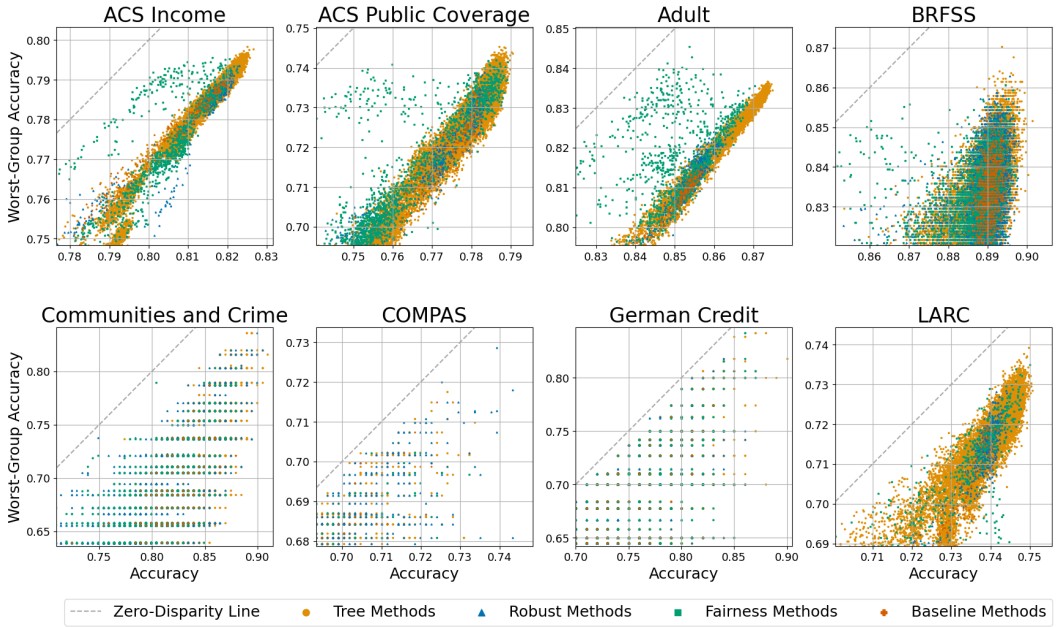

Figure 2: Overall Accuracy vs. Worst-Group Accuracy of robust, fairness-enhancing, tree-based, and baseline models over eight tabular datasets. Dashed lines indicate $y = x$, when worst-group accuracy equal to overall accuracy (zero accuracy disparity). See Figures 12-15 for more detailed results by algorithm. Note: "discretization" artifacts are due to small test/dataset sizes.

A summary of our results is shown in Figures 1, 2, 3, and 4. Following previous works, our main evaluation metrics are accuracy-based: overall accuracy, worst-group accuracy, and accuracy disparity. Over the wide sweep of datasets and model configurations evaluated, modern tree-based models – GBM, LightGBM, Random Forest and XGBoost – all achieve subgroup performance characteristics on par with, or better than, the set of robust or fairness-enhancing models evaluated. For example, on three of the four largest datasets in our study (ACS Income, ACS Public Coverage, LARC),

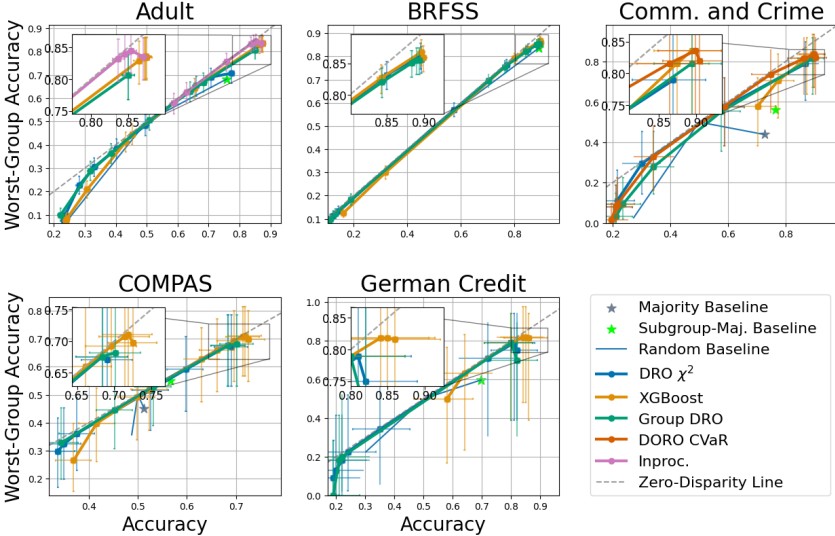

Figure 3: Model performance frontiers, formed by tracing the convex envelope of model performance. Tree-based models achieve comparable and sometimes improved frontiers with the highest-performing robustness methods. (See also Figure 1 for the remaining three datasets.)

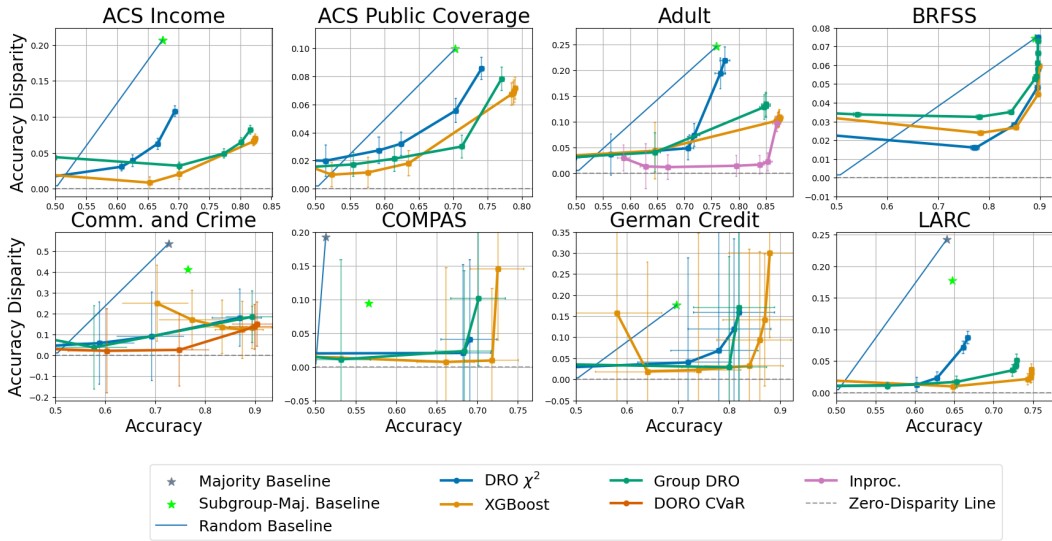

Figure 4: Model disparity frontiers, formed by tracing the convex envelope of model disparity. Tree-based models achieve comparable and sometimes improved frontiers compared to the highest-performing robustness methods, particularly in high-accuracy regions.

XGBoost achieves significantly *better* $\mathcal{L}$ and $\mathcal{L}_w g$ than DRO-based methods (Group DRO, $\chi^2$ DRO), as indicated by nonoverlapping Clopper-Pearson confidence intervals ($\alpha = 0.05$).

The only cases where another algorithm achieves *better* maximum overall robustness (as measured by worst-group accuracy) than tree-based methods are DORO CVaR on Adult, and Inprocessing on Communities and Crime (see Figures 4, 4). We note that in both cases, $(i)$ these differences are not statistically significant, based on the shown Clopper-Pearson confidence intervals at $\alpha = 0.05$, and $(ii)$ these differences occur at points *below* the maximum overall accuracy for each model. In all cases, at the point of maximum overall accuracy, tree-based models achieve equivalent or better worst-group accuracy than all other models evaluated, based on Clopper-Pearson CIs at $\alpha = 0.05$.

We note that this is particularly surprising because none of the tree-based models explicitly optimize for robustness, fairness, or subgroup performance in any way; in contrast, the distributionally robust learners and fairness-enhancing techniques explicitly optimize for such criteria and in some cases (DORO, DRO, Group DRO) provide explicit guarantees of various forms of robustness. A similar analysis showing accuracy *disparity* is shown in Figures 16, 17, and a complete set of model performance observations from each algorithm and dataset, are in Supplementary Section G.

To summarize our results over the large hyperparameter sweeps, we use *model performance frontiers* to measure the best possible set of tradeoffs between ($\mathcal{L}$ and $\mathcal{L}_{\text{WorstGroup}}$) and ($\mathcal{L}$ and $\mathcal{L}_{\text{DISP}}$). Model performance frontiers represent the envelope of the convex hull of all observations of $\mathcal{L}(f_\theta) \in \mathcal{F}$. An example is shown in Figure 1b and 1d, with the remaining datasets in Figures 3 and 4. The frontiers allow us to compare the best-possible tradeoff between accuracy and worst-group accuracy (Fig 3; higher is better) or between accuracy and accuracy disparity (Fig 4; lower is better).

## 5   Results: Robust Models Can Be Metric-Brittle

Our results show that, over all models and datasets, metrics which we refer to as "complementary" – those measuring the same event on different subsets, or with different conditioning – are strongly correlated with each other: accuracy and worst-group accuracy; CVaR and CVaR DORO; and Demographic Parity Difference and Equalized Odds Difference all show strong correlation with each model class $\mathcal{F}$; we show these correlations on Adult in Figure 5. The median $\rho$ values over all datasets and algorithms are 0.87, 0.99, 0.79 for the three pairs of metrics, respectively (we show the complete set of correlations for each model and metric in Figure 18). These results show that model selection based on one metric from each pair (Overall Accuracy) is likely to lead to a model with strong performance for the other metric in the pair (e.g. Worst-Group Accuracy).

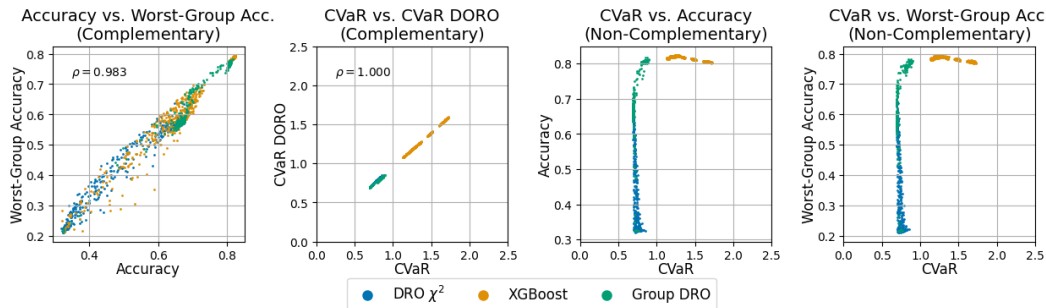

Figure 5: While "complementary" metrics (those measuring the same event with different conditioning) are closely correlated for all models, non-complementary metrics behave differently by model. Accuracy can vary widely for "robust" models with a fixed robust (CVaR) risk, while for tree models, the best (lowest) CVaR risk is typically associated with the best (highest) accuracy. ACS Income dataset shown; see Supplementary Figure 18 and Section G for additional datasets and metrics.

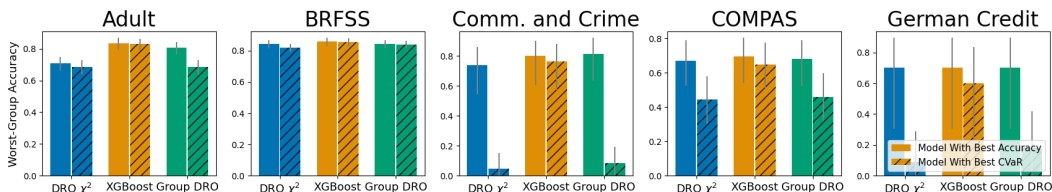

Figure 6: Worst-group accuracy for models with best overall accuracy (solid), and best CVaR (shaded). Tree-based models show better performance across non-complementary metrics (as indicated by similar heights of orange bars within each plot), whereas robustness- and fairness-based models do not (statistically-significant gaps between blue, green bars): worst-group accuracy drops significantly between best-accuracy and best-CVaR DRO $\chi^2$ and Group DRO models (See also Figure 1 for the remaining three datasets.).

In contrast, our experiments show that there is a severe lack of correlation across almost all of the non-complementary pairs, shown in the bottom row of Figure 5. In particular, our results show that models which achieve a strong "robust" risk – for example, a low CVaR DORO risk, one model selection rule used in [74] – can achieve very low accuracies. Indeed, the "robust" learning methods are the most susceptible to this discrepancy over metrics in our experiments, as shown in Figure 6. While these results confirm the finding in previous works that fairness metrics are correlated [26], our broader comparison shows a brittleness to model selection metrics outside these sets of complementary metrics, and reveal the very strict sense in which robustness guarantees apply only to a limited range of metrics. This discrepancy is of practical significance given that robust models are often selected using robust metrics (i.e. DORO CVaR, cf. [74]), while models in practice are frequently evaluated using other metrics (i.e. accuracy, fairness metrics).

To illustrate the practical implications of this metric brittleness, we also explore the impact of various strategies for choosing a single $f_\theta \in \mathcal{F}$ over a set of models in each hyperparameter grid. While this problem is implicitly addressed in almost all previous works, the decision is often handled differently. For example, models are tuned and selected by hand [46], chosen based on worst-group performance [74], or based on robust risk metrics [62]. The empirical implications of model selection methods for fairness and robustness are not well-understood.

To address this question, we show the worst-group performance of models over our sweeps, selected according to either *best accuracy* or *best CVaR* in Figure 6. Figure 6 demonstrates the downstream impact of the lack of correlation between robust risk metrics and worst-group accuracy: distributionally-robust models can suffer significant drops in worst-group accuracy, when selected based on robust risk. In contrast, tree-based models show "metric robustness" – that is, tree-based models which perform best according to the robust risk measure (CVaR) still achieve worst-group accuracy near the highest-accuracy model of the same class (here, XGBoost).

# 6 Results: Trees Show Lower Hyperparameter Sensitivity and are Less Costly to Train

For many practical applications, both the time and financial costs of training models are of prime importance. These also affect the amount of expertise required to train a model (with sensitive models requiring greater expertise) and directly impact the carbon footprint of training [34]. We conduct a hyperparameter sensitivity analysis by pruning the hyperparameter search space to eliminate configurations that performed poorly across all datasets, ranking the remaining configurations by performance (i.e. accuracy), and plotting the decline in performance over the ranked models. These results, shown in Figure 7 and Section F.2, demonstrate that tree-based models also show less sensitivity to hyperparameters than the other methods evaluated for both overall performance metrics (e.g., accuracy) and subgroup robustness metrics (e.g., worst-group accuracy).

These results are particularly important in light of the large differences in resources required to achieve similar levels of performance between robust models and the tree-based models: for example, the full hyperparameter grid sweep of size $12k$ XGBoost on the largest dataset in our study, ACS Income, completed in 1 CPU-day; DRO $\chi^2$ sweep of 3250 training runs completed in 58 *GPU*-days. Due to the differences in hardware required to train various models, we conduct an estimated comparison of the *cost* of training an individual model, and of the full sweep. These results, shown in Figure 19, show that tree-based models are also considerably less expensive to train and tune.

# 7 Conclusion

Machine learning has made significant progress in the past several years in identifying and addressing various challenges which can contribute to real-world performance disparities. However, our results suggest that another form of progress not widely acknowledged as having implications for fairness or robustness – advances in tabular data modeling with tree-based algorithms – have similar, if not greater, impact in practice than the state of the art in robust neural network learning or fairness-enhancing methods. Our results suggest that tree-based algorithms are a surprisingly strong baseline for subgroup robustness in tabular data, and that future work should compare against, and improve upon, the subgroup robustness of tree-based models. Our work suggests that, in practice, tree-based ensembles are an effective default for tasks where subgroup robustness is desired.

Our results also suggest that subgroup robustness on par with existing state of the art can be achieved with tree-based classifiers that are easier and computationally cheaper to train and tune than either robust or fairness-enhancing models, which tend to scale poorly with dataset size (in both $n$ and $d$). Our work thus contributes a critical baseline, similar to how other recent works have contributed much-needed baselines in areas of machine learning affected by the rapid proliferation of new methods [59, 67, 47], and provides a strong benchmark for future robust and fair learning methods.

Our findings are limited to the set of hyperparameters and models explored in our experiments – a superset of those from the works discussed above (e.g. [62, 74]). Our findings do not demonstrate that robust learning methods *cannot* achieve subgroup robustness on par with e.g. XGBoost, but merely that this is not achieved with the configurations widely used in the literature. We note that the loss-based interventions largely favored by existing robust and fair learning techniques require optimization over a continuous function, typically implemented as an MLP; this also makes it difficult to disentangle whether the functional form, or the training procedure and objective, lead to the improved subgroup robustness of tree-based models observed in this study. Future work should investigate the subgroup robustness of non-MLP models trained using robust techniques.

While our experiments do not identify the *cause* of trees' subgroup robustness, it is likely that this is a consequence of their strong overall performance on tabular data. These findings can be also viewed as an analogue of the empirical relationship between in-distribution and out-of-distribution accuracy in computer vision documented in [49] but now demonstrated for the tabular domain. It is possible that these improvements are due to $(i)$ an inductive bias in tree-based models beter suited to modeling differences in subpopulations in tabular data, or $(ii)$ the *ensembling* used by the tree-based models in this work. We leave an identification of such causal factors to future work.

# 8 Acknowledgements

This work is in part supported by the NSF AI Institute for Foundations of Machine Learning (IFML, CCF-2019844), Open Philanthropy, Google, and the Allen Institute for AI.

Moreover, our research utilized computational resources and services provided by the Hyak computing cluster at the University of Washington, and by Advanced Research Computing at the University of Michigan, Ann Arbor.

We are also grateful to Hongseok Namkoong, Tatsunori Hashimoto, John Miller, Michael Kim, Shafi Goldwasser, and attendees of the 2022 Institute for Foundations of Data Science Workshop on Distributional Robustness for useful feedback and discussions about the work, and to Christopher Brooks for assistance with the Learning Analytics Architecture (LARC) dataset.

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
