Figure 7: Performance over (truncated) hyperparameter grids on all datasets. Tree-based models (orange) show significantly less sensitivity to hyperparameter settings, for both subgroup and overall performance metrics (worst-group accuracy shown here), as indicated by nonoverlapping Clopper-Pearson CIs ($\alpha = 0.05$). For additional results and methodology for constructing these plots, see Section F.2.

## A    Dataset Details

This section provides further detail on the datasets used in this work, along with their preprocessing.

For each dataset, we use an 80%/10%/10% train/validation/test split. The only exceptions are ACS datasets and LARC, where we use equally-sized train/test/validation splits (see below), and Adult, where we use the official train-test split.

For each dataset, we use two binary sensitive attributes derived from existing features. These attributes are primarily selected to both align with real-world sensitive attributes in practice, and, where possible, to match the implementations in previous works. For methods which use group information, each intersection of the sensitive attributes are considered a separate group (for a total of $2^2$ nonoverlapping subgroups).

- **ACS Income[2]:** Proposed in [20], consists of approximately 160k responses to the 2018 American Community Survey. Since it is proposed as a replacement for the Adult dataset, we perform income prediction task, where the label is an indicator for whether an individuals' income exceeds the median ($56,000$). Sensitive attributes are race and gender. We use a larger set of features than that described in [20], which we found to improve classifier performance. The sensitive feature for race is coded as "white alone" vs. other categories, as in [20]. We use a subsample of the full 1.6M records for our experiments due to computational constraints (for example, fairness methods scale linearly or quadratically in dataset size and feature size; robustness methods also incur extra costs as data dimensionality increases).

- **ACS Public Coverage:** Derived from the same raw data source as ACS Income, described above. The task is to predict whether an individual is covered by public health insurance. Sensitive attributes are race and gender. We use identical feature set to [20]. The sensitive features and subsampling are handled as in ACS Income.

- **Adult[3]:** A widely-used benchmark dataset derived from 1994 US Census data [45]. The task is to predict whether an individuals' income exceeded $50,000$. Sensitive attributes are

[2] https://github.com/zykls/folktables
[3] https://archive.ics.uci.edu/ml/machine-learning-databases/adult/

race and gender. We use the standard train-test split for Adult, following the preprocessing code of [6][4], and we further split the set partition evenly into validation and test.

- **BRFSS:** The Behavioral Risk Factors and Surveillance System[5] is a large-scale phone survey conducted annually from a random sample of adults in the United States by the Centers for Disease Control and Prevention [6]. The objective of the BRFSS is to collect uniform, state-specific data on preventive health practices and risk behaviors that are linked to chronic diseases, injuries, and preventable infectious diseases in the adult population. Respondents answer questions related to personal health, lifestyle habits, and health care coverage. BRFSS completes more than 400,000 adult interviews each year, making it the largest continuously conducted health survey system in the world. We use the BRFSS sample from 2015.

- **Communities and Crime[7]:** A set of features describing a community, and the prediction target is whether the community has an elevated crime rate. Following [43, 41, 42], we predict a binary label for whether the violent crime rate exceeds a threshold of 0.08. Sensitive attributes are race and income level. We use the preprocessing of [43] for this dataset, where the race feature is a binary indicator for whether the feature `racePctWhite` $> 0.85$ and the income level is an indicator for whether the income is above the median community income.

- **COMPAS[8]:** Parole records from Florida, USA, where the task is to predict whether an individual will recidivate within two years. Sensitive attributes are race and gender.

  While every labeled dataset contains human biases inherent in collecting and categorizing the data, the COMPAS dataset in particular has been the subject of valid critiques, and is mostly included for comparisons to prior work. We note that the COMPAS dataset reflects the patterns of policing and social processes in a particular community (South Florida) at a specific time, and as others have noted [69], each stage of the COMPAS dataset's creation introduces opportunities for bias [60, 4] and that measurement biases and errors with COMPAS have been widely documented [3].

- **German Credit[9]:** Credit application records, where the goal is to predict whether an individual has low or high credit risk. Sensitive attributes are age and gender. There are two versions of the German Credit dataset, a "numeric" version which contains binarized versions of most categorical features (with some removed), and a non-numeric version, which also contains categorical features. We do *not* use the "numeric" version of the dataset used by several other works. We found the numeric version of the dataset to be poorly-documented, lack useful features which were present in the "non-numeric" version, and contain features which actually mixed multiple variables . Using the non-numeric version, we extract separate features for sex and marital status (which are combined under a single feature in the numeric dataset).

- **LARC[10]:** The Learning Analytics Data Architecture (LARC) Data Set is a research-focused data set containing information about students who have attended the University of Michigan since the mid-1990s. The data includes features related to students, their enrollment, and performance, similar to the electronic records stored by many institutions of higher education. The data is divided into that which is constant throughout a student's academic career (e.g., ethnicity, SAT test scores, high school GPA, and earned degrees), that which can change from term to term (e.g., academic level, academic career, term GPA, and enrolled credits), and that which can change from class to class (e.g., subject, catalog number, earned grade, etc.). The prediction target is an indicator for whether a student will receive a grade above the median in a course; this is commonly referred to as "at-risk" grade prediction and is used to identify students at risk of struggling in a course. Sensitive attributes are "Underrepresented Minority" status (a common indicator of diversity reported by all accredited institutions in the United States) and students' self-reported sex.

---

[4]https://fairmlbook.org/code/adult.html
[5]https://www.kaggle.com/datasets/cdc/behavioral-risk-factor-surveillance-system
[6]https://www.cdc.gov/brfss/annual_data/annual_data.htm
[7]https://archive.ics.uci.edu/ml/datasets/communities+and+crime
[8]https://github.com/propublica/compas-analysis
[9]https://archive.ics.uci.edu/ml/datasets/statlog+(german+credit+data)
[10]https://enrollment.umich.edu/data/learning-analytics-data-architecture-larc

It is critical to note that any notion of fairness or robustness in real-world societal contexts involves much more than even the sets of two demographic attributes considered here [68]. We also note that our treatment of these sensitive attributes as binary, while consistent with the majority of the fairness and robustness literature upon which this work is based, is necessarily reductionistic, and in practice many of these sensitive attributes are neither binary [63] nor fixed [30].

Furthermore, we urge readers to consider that, while these datasets are commonly used as benchmarks for fair or subgroup-learning, there are important social and contextual factors that must be considered for any real-world deployment of a model in these tasks to be considered "fair" [3].

# B   Model Details

**Fairness-Enhancing Models:** Following [20], we evaluate one method each of pre-, in-, and postprocessing. The preprocessing method of [73] attempts to learn a transformation of the original inputs which minimally distorts the original data and its relationship to the labels, while ensuring that both group fairness (the proportion of members in a protected group receiving positive classification is identical to the proportion in the overall population) and individual fairness (similar individuals are treated similarly). The inprocessing method of [1] attempts to reduce fair classification to a cost-sensitive classification problem, where the goal is to minimize the prediction error subject to one or more fairness constraints. Finally, the postprocessing method we use is from [32], which randomizes the predictions of a fixed $f_\theta$ to satisfy equalized odds criterion. All fairness methods are implemented to simultaneously satisfy their constraints across both sensitive attributes in our datasets. We use the implementations of `aif360` [9] and `fairlearn` [12] for all fairness interventions.

None of the fairness methods encompasses a prediction model, so following [20], we pair each method (in-, pre-, and postprocessing) with a gradient-boosted tree (GBM) [27, 51]. However, unlike [20], we perform hyperparameter sweeps for both the fairness methods *and* the GBM.

**Distributionally Robust Models:** We draw from several classes of modern robust learning techniques. We utilize two variants of Distributionally Robust Optimization (DRO), which attempts to solve

$$\min_\theta \sup_{Q \in \mathcal{U}(D)} \mathbb{E}_{S \sim Q}\big[\mathcal{L}(f_\theta)\big] \tag{3}$$

where $\mathcal{U}$ defines an uncertainty set with respect to the training distribution $D$. We evaluate two widely-used formulations of the uncertainty set $\mathcal{U}$ in Equation (3). The first is the set of distributions with bounded likelihood ratio to $D$, such that (3) defines the conditional value at risk (CVaR). The second is the set of distributions with bounded $\chi^2$-divergence to $D$. We refer to these as CVaR-DRO and $\chi^2$-DRO, respectively. We use the efficient implementation of [46] for these methods.

We also evaluate the "Distributional and Outlier Robust Optimization" (DORO) of [74], which adds an $\epsilon$ parameter to (3) such that only the $\epsilon$-smallest fraction of the data, when sorted by $\mathcal{L}$, are considered at each update step; this has the effect of ignoring outliers.

We additionally evaluate Group DRO [62], seeks to minimize the worst-group loss by solving

$$\min_\theta \sup_{g \in \mathcal{G}} \mathbb{E}_{(x,y) \sim \mathcal{D}_g}\big[\mathcal{L}(f_\theta)\big] \tag{4}$$

. Finally, we evalute the Maximum Weighted Loss Discrepancy (MWLD) of [43]. This formulation adds an extra term, $\mathcal{L}_{\mathrm{LV}} \coloneqq \mathrm{Var}(\mathcal{L}(f_\theta(x_i) \in X)$, to the ERM objective during training. The MWLD objective is shown in [43] to be related to group fairness and robustness to subgroup shifts.

**Tree-Based Models:** As we note in Section 2.3, several modern tree-based methods achieve effectively identical performance on many tasks, with several flavors of gradient-boosted trees (GBM, LightGBM, CatBoost, XGBoost) widely being considered the state-of-the art models for tabular data. Therefore, we evaluate GBM (in order to compare directly to [20], which combines GBM with our fairness methods of interest) and LightGBM (due to its scalability, which is required for large-scale hyperparameter tuning over the datasets in this work). We also evaluate Random Forests in order to compare to a non-gradient-boosted tree-based classifier.

**Baseline Supervised Learning Models:** In addition to the above-described methods, we also include the following standard supervised learning methods for comparison: $L_2$-regularized logistic regression, and Support Vector Machines (SVM). For the SVM methods, because learning nonlinear kernels for large datasets with many features can be prohibitively expensive, we instead use two kernel approximation methods for learning: the Nystroem kernel method [23, 70] and random Fourier features [57, 58, 71]. For all baseline methods, we use the implementation of [52].

# C   Additional Metrics

In addition to the metrics reported and defined in Section 3.4, we also use the following metrics in our supplementary results reported below:

**Accuracy:** The accuracy is defined as the fraction of labels correctly predicted at a given threshold: $\mathcal{L}_{\text{Acc}} \coloneqq \mathbb{1}(\hat{y} == y)$, where $\hat{y} = \mathbb{1}(f_\theta(x) >\geq t)$ is the predicted label of $x$ using threshold $t$. We use $t = 0.5$ throughout.

**Cross-Entropy:** We use the standard binary cross-entropy measure, defined as $\mathcal{L}_{\text{ce}}(f_\theta; x, y) = -y \log(f_\theta(x)) - (1 - y) \log(1 - f_\theta(x))$.

**DORO CVaR Risk:** The DORO CVaR risk is a version of CVaR risl (2) which excludes the $\epsilon$-largest-loss elements in $D$ in an effort to avoid outliers. Formally, the DORO CVaR risk is:

$$\mathcal{L}_{\text{CVaR-DORO}}(D, \mathcal{P}, \epsilon) \coloneqq \inf_{D'} \mathcal{L}_{\text{CVaR}}(D, \mathcal{P}') : \exists \tilde{\mathcal{P}}' \quad \text{s.t.} \quad \mathcal{P} = (1 - \epsilon)\mathcal{P}' + \epsilon \tilde{P}' \tag{5}$$

where $\epsilon$ is a hyperparameter corresponding to the fraction of outliers in the dataset. We note that (5) is the loss function directly optimized by the DORO-CVaR model, but it has been used as a more general evaluation of the outlier-robust tail risk of a classifier (cf. [74]).

**Demographic Parity Difference:** Demographic Parity (DP) is a fairness criterion that indicates the positive prediction rates across two disjoint subgroups $a, a' \in A$ are equal [6]. That is, when demographic parity is satisfied, $P(f_\theta(x_i; a_i = a) = 1) = P(f_\theta(x_j; a_j = a') = 1)\forall i, j$. The Demographic Parity Difference measures the degree to which this constraint is violated, and for nonbinary sensitive subgroups, it measures the worst-case difference:

$$\mathcal{L}_{DP-Diff} \coloneqq \max_{a,a'} \left| P(f_\theta(x_i; a_i = a) = 1) - P(f_\theta(x_j; a_j = a') = 1) \right| \tag{6}$$

**Equalized Odds Difference:** Equalized Odds (EO) is a fairness criterion indicating that the true positive and false positive rates are equal across two groups. The Equalized Odds Difference measures worst-case violation of Equalized Odds, across sensitive subgroups (this is $\mathcal{L}_{DISP}$ with $\mathcal{L}$ as DP). Equalized Odds Difference can be formulated as the greater of two metrics: the true positive rate difference, and the false positive rate difference [12].

# D   Model Performance Frontier Curves

This section describes how Model Performance Frontier curves are computed. We note that this work is not the first to use convex envelopes as a way to understand model performance; for example, [1] uses convex envelopes to understand the relationship between fairness constraint violations and model error.

To compute a model envelope in 2D, we use an algorithm to compute the convex hull, and then trace the relevant edge of the convex hull corresponding to the best-achieved performance tradeoffs under the two metrics (depending on whether these metrics are maximized, or minimized).

We provide Python code to compute these curves in the code repository associated with this paper; for completeness, we also provide the full algorithm in Algorithm 1.

**Algorithm 1** Model performance frontiers. This shows the computation where higher values are better for both metrics $m^{(1)}, m^{(2)}$.

---

**Input:** $\left(m_i^{(1)}, (m_i^{(2)})\right)_{i=i}^{|\mathcal{G}|}$ ▷ Model performance metrics $m^{(1)}, m^{(2)}$ for each configuration in grid $\mathcal{G}$
**Input:** ConvexHull, a method which computes the convex hull for a set of points and returns them in clockwise order.
**Output:** Idxs $\subseteq 1, \ldots, |\mathcal{G}|$           ▷ Indices of inputs on frontier.
    vertices $\leftarrow$ ConvexHull(Input)
    idxs $\leftarrow$ [ ]           ▷ Initialize array of frontier points.
    top_idx $= \text{argmax}_{m_+i^{(2)}}\left(m_i^{(1)}, (m_i^{(2)})_i \in \text{vertices}\right.$       ▷ Add uppermost point to frontier
    idx $\leftarrow$ top_idx
    idxs $\leftarrow$ idxs $+$ [idx]
    $m_{curr}^{(1)}, m_{curr}^{(2)} \leftarrow$ vertices[idx]
    next $\leftarrow (|\text{vertices}| + 1) \bmod |\text{vertices}|$
    $m_{next}^{(1)}, m_{next}^{(2)} \leftarrow$ vertices[next]
    **while** $m_{next}^{(1)} < m_{curr}^{(1)}$ **do**           ▷ Trace frontier counterclockwise
       idxs $\leftarrow$ next
       idx $\leftarrow$ (idx $+ 1$) mod |vertices|
       next $\leftarrow$ (next $+ 1$) mod |vertices|
       $m_{curr}^{(1)}, m_{curr}^{(2)} \leftarrow$ vertices[idx]
       $m_{next}^{(1)}, m_{next}^{(2)} \leftarrow$ vertices[next]
    **end while**
    idx $\leftarrow$ top_idx
    next $\leftarrow (|\text{vertices}| - 1) \bmod |\text{vertices}|$
    $m_{curr}^{(1)}, m_{curr}^{(2)} \leftarrow$ vertices[idx]
    $m_{next}^{(1)}, m_{next}^{(2)} \leftarrow$ vertices[next]
    **while** $m_{next}^{(1)} > m_{curr}^{(1)}$ **do**           ▷ Trace frontier clockwise
       idxs $\leftarrow$ next
       idx $\leftarrow$ (idx $- 1$) mod |vertices|
       next $\leftarrow$ (next $- 1$) mod |vertices|
       $m_{curr}^{(1)}, m_{curr}^{(2)} \leftarrow$ vertices[idx]
       $m_{next}^{(1)}, m_{next}^{(2)} \leftarrow$ vertices[next]
    **end while**
    **return** idxs

---

# E   Training Details

We train all models using the original optimizer, SGD, used in their original works [74, 46, 43, 62]. For all models, we train for a fixed number of epochs, but keep the weights from the best epoch based on the loss on the validation set. The number of epochs used for each dataset is as follows: ACS Income 50 epochs; Adult 300 epochs; Communities and Crime 100 epochs; COMPAS 300 epochs; German 50 epochs.

For all neural network-based models, we used a fixed batch size of 128 as in [38]; we found that varying the batch size did not affect performance but significantly increased the computational cost of hyperparameter sweeps. This also ensures that each model trained with batching sees the same number of examples during training on a given dataset.

Neural-network-based models were trained on GPU, either NVIDIA RTX 2080 Ti GPUs with 11GB of RAM, or NVIDIA Tesla M60 GPUs with 8 GB of RAM.

# F   Hyperparameter Grids

## F.1   Grid Definition

We detail the hyperparameter grids for each experiment in Table 2. For each dataset, we perform a full hyperparameter grid sweep.

For methods which are built on a "base" model (i.e. Group DRO, which uses an MLP model, or Preprocessing, which is paired with GBM as in [20]), we tune the full grid of hyperparameters for the base model in addition to the hyperparameters for that method, as indicated in Table 2. We note that this is not always the case in previous works; for example, [20] uses the default hyperparameters for GBM with fairness methods, and many DRO-based works use a fixed architecture or optimization hyperparameters for their published comparisons.

We use default parameters with the given implementation for all methods except where indicated.

**Tree-based methods:** For GBM and random forest, we use the implementation of [52]. For Light-GBM we use the original implementation of [40][11]. For XGBoost we use the original implementation of [16][12]. For each method, we construct hyperparameter grids by beginning with large sweeps around the default hyperparameters of each method, and then pruning the sweeps to a tractable size manually by inspecting accuracy, robustness, and fairness metrics. For XGBoost, we only use training methods available with GPU-based training to ensure scalability (note that only CPU-based training was used for XGBoost models in our experiments).

**Robustness-enhancing methods:** For robustness-enhancing methods, our hyperparameter grids combine our large default MLP grid with the hyperparameter grids for method-specific parameters used in previous works (e.g. [74, 46]). We use the implementation of [74] for DORO and [46] for DRO methods. Note that we do not conduct sweeps for DORO models on the Adult and ACS datasets due to computational limitations, as full sweeps for both methods are prohibitively expensive to run on these large datasets.

**Fairness methods:** We use standard implementations and the largest-possible grids while meeting our computational constraints, as many of the fairness methods scale worse than linearly with dataset size, feature size, or both. For LFR, we fix Ax as in [73], and otherwise use the center portion of the grid from [20] which we found to be sufficient for our sweeps when also tuning the GBM parameters (which was not performed in [20]) across our datasets. Our grid for inprocessing uses the same constraints explored in [20] but also tunes the constraint slack and GBM parameters, which were not tuned in [20]. We use the implementation of [8] for LFR and postprocessing, and [12] for inprocessing.

## F.2   Hyperparameter Sensitivity Analysis

This section provides exploratory results regarding the *sensitivity* of the models evaluated to hyperparameter configurations. For each model, we take the full set of hyperparameter configurations evaluated. then for each hyperparameter, we compute the set of values of the best-performing model (here, using worst-group accuracy) over all datasets, dropping values from continuous hyperparameter grids outside this range. This truncation step eliminates ranges of each hyperparameter which performed poorly for *all* datasets. Finally, we order the remaining hyperparameter configurations by worst-group accuracy to construct the plots below.

In the top panel of Figure 8, we show only the DRO $\chi^2$, XGBoost, and Group DRO models, following our running example in the main text. We include 95% Clopper-Pearson confidence intervals using the *smallest* sensitive subgroup as the sample size for the CI.[13] In the bottom panel of Figure 8, scale the results according to the *best* performance achieved by each model class on the target dataset.

We provide similar results in the top and bottom rows of Figure 9 which include all models; here we omit the confidence intervals due to space (although the intervals would have the same width as in Figure 8).

---

[11]https://github.com/microsoft/LightGBM

[12]https://github.com/dmlc/xgboost

[13]This makes the confidence intervals in 8 conservative, as they would be narrower when the worst group is not the smallest group; we do this so that the interval width is consistent for all model configurations.

| Model | Grid Size | Hyperparameter | Values |
|---|---|---|---|
| **Baseline Methods** | | | |
| ♣ MLP | 405 | Learning Rate | $\{1e^{-1}, 1e^{-2}, 1e^{-3}, 1e^{-4}, 1e^{-5}\}$ |
| | | Weight Decay | $\{0, 0.1, 1\}$ |
| | | Num. Layers | $\{1, 2, 3\}$ |
| | | Hidden Units | $\{64, 128, 256\}$ |
| | | Momentum | $\{0., 0.1, 0.9\}$ |
| | | Batch Size | $\{128\}$ |
| SVM | 576 | C | $\{0.01, 0.1, 1., 10., 100., 1000.\}$ |
| | | Kernel Appx. | $\{\text{Nystroem, RKS}\}$ |
| | | Loss | Squared Hinge |
| | | $\gamma$ | $\{0.5, 1.0, 2.0\}$ |
| | | Num. Components | $\{64, 128, 256, 512\}$ |
| | | Nystroem Kernel Degree | $\{2, 3\}$ |
| | | Nystroem Kernel | $\{\text{RBF, poly}\}$ |
| Logistic Regression | 8 | $L_2$ penalty | $\{0.001, 0.01, 0.1, 1., 10., 100., 1000., 10000.\}$ |
| **Tree-Based Methods** | | | |
| ♢ GBM | 100 | Learning Rate | $\{0.01, 0.1, 0.5, 1.0, 2.0\}$ |
| | | Num. Estimators | $\{64, 128, 256, 512, 1024\}$ |
| | | Max Depth | $\{2, 4, 8, 16\}$ |
| | | Min. Samples Split | 2 |
| | | Min. Samples Leaf | 1 |
| Random Forest | 640 | Num. Estimators | $\{64, 128, 256, 512\}$ |
| | | Max Features | $\{\text{sqrt, log2}\}$ |
| | | Min. Samples Split | $\{2, 4, 8, 16\}$ |
| | | Min. Samples Leaf | $\{1, 2, 4, 8, 16\}$ |
| | | Cost-Complexity $\alpha$ | $\{0., 0.001, 0.01, 0.1\}$ |
| XGBoost | 1944 | Learning Rate | $\{0.1, 0.3, 1.0, 2.0\}$ |
| | | Min. Split Loss | $\{0, 0.1, 0.5\}$ |
| | | Max. Depth | $\{4, 6, 8\}$ |
| | | Column Subsample Ratio (tree) | $\{0.7, 0.9, 1\}$ |
| | | Column Subsample Ratio (level) | $\{0.7, 0.9, 1\}$ |
| | | Max. Bins | $\{128, 256, 512\}$ |
| | | Growth Policy | $\{\text{Depthwise, Loss Guide}\}$ |
| LightGBM | 12544 | Learning Rate | $\{0.01, 0.1, 0.5, 1.\}$ |
| | | Num. Estimators | $\{64, 128, 256, 512\}$ |
| | | $L_2$-reg. | $\{0., 0.00001, 0.0001, 0.001, 0.01, 0.1, 1.\}$ |
| | | Min. Child Samples | $\{1, 2, 4, 8, 16, 32, 64\}$ |
| | | Max. Depth | $\{\text{None}, 2, 4, 8\}$ |
| | | Column Subsample Ratio (tree) | $\{0.4, 0.5, 0.8, 1.\}$ |
| **Robustness-Enhancing Methods** | | | |
| DORO $\chi^2$ ♣ | 12150 | Uncertainty set size $\alpha$ | $\{0.1, 0.2, 0.3, 0.4, 0.5, 0.6\}$ |
| | | Outlier proportion $\epsilon$ | $\{0.001, 0.01, 0.1, 0.2, 0.3\}$ |
| DORO CVaR ♣ | 12150 | Uncertainty set size $\alpha$ | $\{0.1, 0.2, 0.3, 0.4, 0.5, 0.6\}$ |
| | | Outlier proportion $\epsilon$ | $\{0.001, 0.01, 0.1, 0.2, 0.3\}$ |
| DRO $\chi^2$ ♣ | 2835 | Uncertainty set size $\alpha$ | $\{0.01, 0.1, 0.2, 0.3, 0.4, 0.5, 0.6\}$ |
| DRO CVaR ♣ | 2835 | Uncertainty set size $\alpha$ | $\{0.001, 0.01, 0.1, 0.2, 0.3, 0.4, 0.5, 0.6\}$ |
| Group DRO ♣ | 1620 | Group weights step size | $\{0.001, 0.01, 0.1, 0.2\}$ |
| MWLD ♣ | 6075 | $L_2$ penalty | $\{0, 0.1, 1\}$ |
| | | Loss variance penalty | $\{1e^{-3}, 1e^{-2}, 1e^{-1}, 1, 10, \}$ |
| **Fairness-Enhancing Methods** | | | |
| Preprocessing ♢ | 2500 | Ax | 0.01 |
| | | Ay | $0.001, 0.01, 0.1, 1, 10$ |
| | | Az | $0.001, 0.01, 0.1, 1, 10$ |
| Inprocessing ♢ | 1000 | Constraint slack $\epsilon$ | $\{1e^{-4}, 1e^{-3}, 1e^{-2}, 1e^{-1}, 1\}$ |
| | | Constraint Type | $\{\text{DP, EO}\}$ |
| | | Max Iter. | 200 |
| Postprocessing ♢ | 100 | No tunable hyperparameters | |

Table 2: Hyperparameter grids used in all experiments. ♣: all MLP parameters also tuned. ♢: all GBM parameters also tuned.

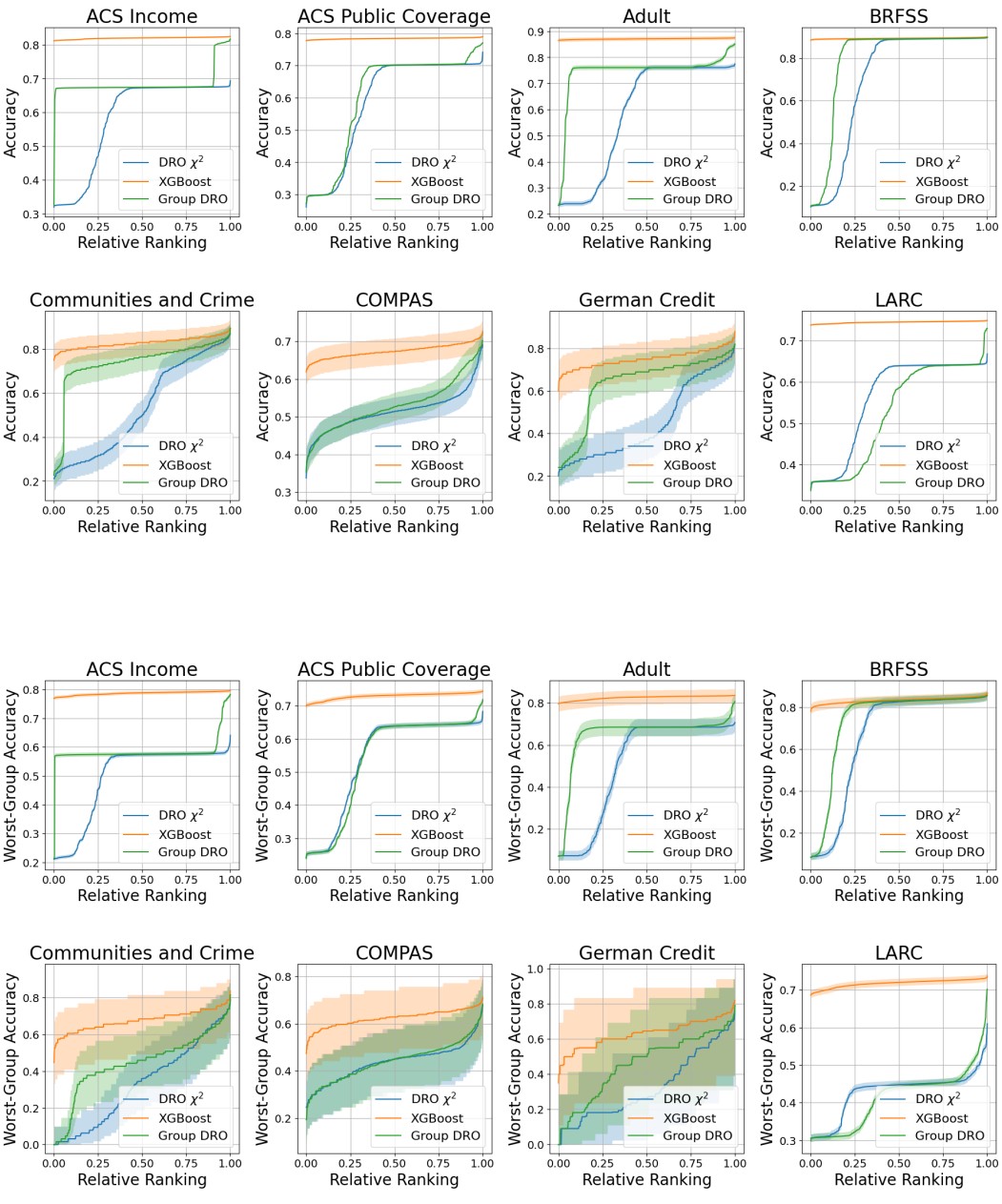

Figure 8: Hyperparameter sensitivity plots for $\chi^2$ DRO, Group DRO, and XGBoost models. The top 8 panels show Accuracy; the lower 8 panels show worst-group accuracy (this is identical to Figure 7, reproduced here for clarity). XGBoost shows considerably lower sensitivity to hyperparameter tuning.

# G   Additional Results

This section contains additional experimental results not included in the main text, along with fine-grained displays of the results summarized in the main figures.

## G.1   Training Compute Cost

In Section 6, we discuss hyperparameter sensitivity and training time of the various algrorithms present in our study. Here, we provide estimations of the result of conducting our hyperparameter

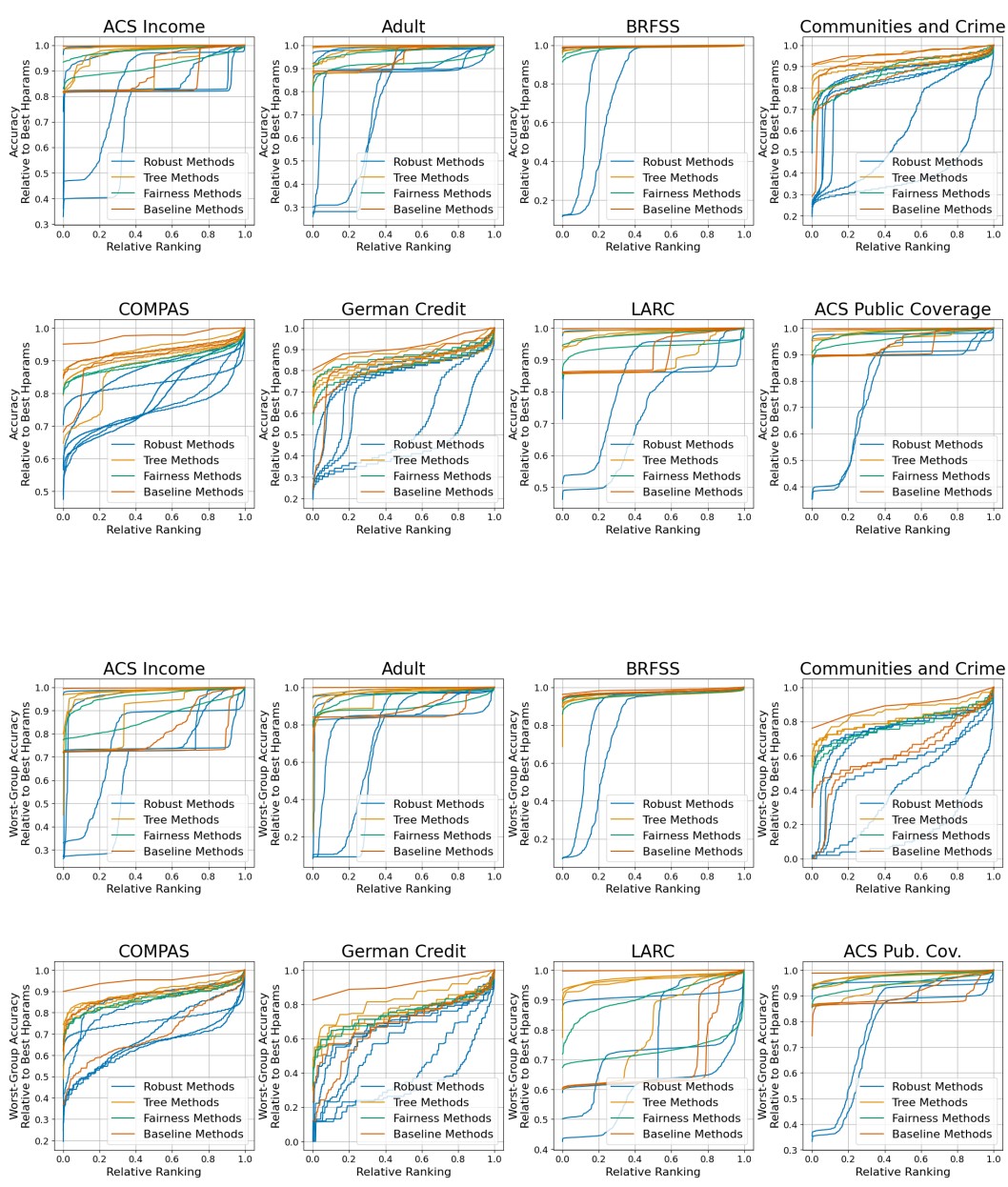

Figure 9: Hyperparameter sensitivity plots for all models evaluated. (Clopper-Pearson CIs omitted due to space).

sweeps on modern cloud-based computing platforms, in order to estimate the costs of individual training runs, and the full hyperparameter sweeps, in our study.

Figure 19 displays the estimated median cost of a single training run of each model in (DRO $\chi^2$, XGBoost, Group DRO, LightGBM). We note that while XGBoost and LightGBM are trained on CPU in this study (although GPU implementations of each training algorithm are available), training of the DRO models at scale is only feasible on GPU.

For our cost estimation, we use prices of \$7.20 per compute-hour for GPU and \$3.072 per compute-hour for CPU, which reflect the hourly price of cloud-based GPU and CPU hardware used in this study.

Figure 19 shows that, compared to DRO methods, tree-based methods (XGBoost, LightGBM) achieve comparable cost or considerable savings for all datasets in our study. The lone exception is XGBoost

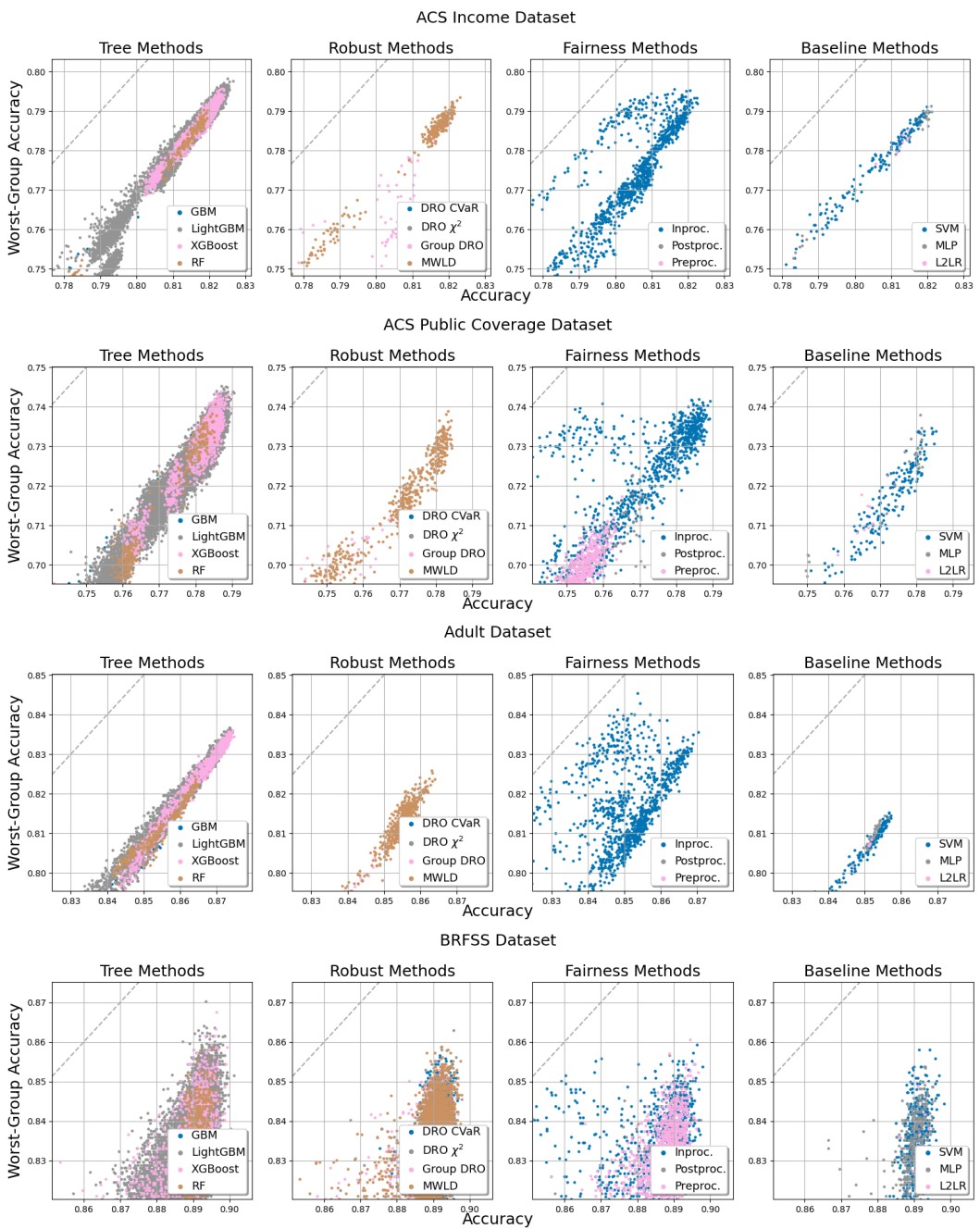

Figure 10: Overall Accuracy vs. Worst-Group Accuracy of robust, fairness-enhancing, tree-based, and baseline models over eight tabular datasets (ACS Income, ACS Public Coverage, Adult, BRFSS). For results by individual algorithm, see Figures 12- 15.

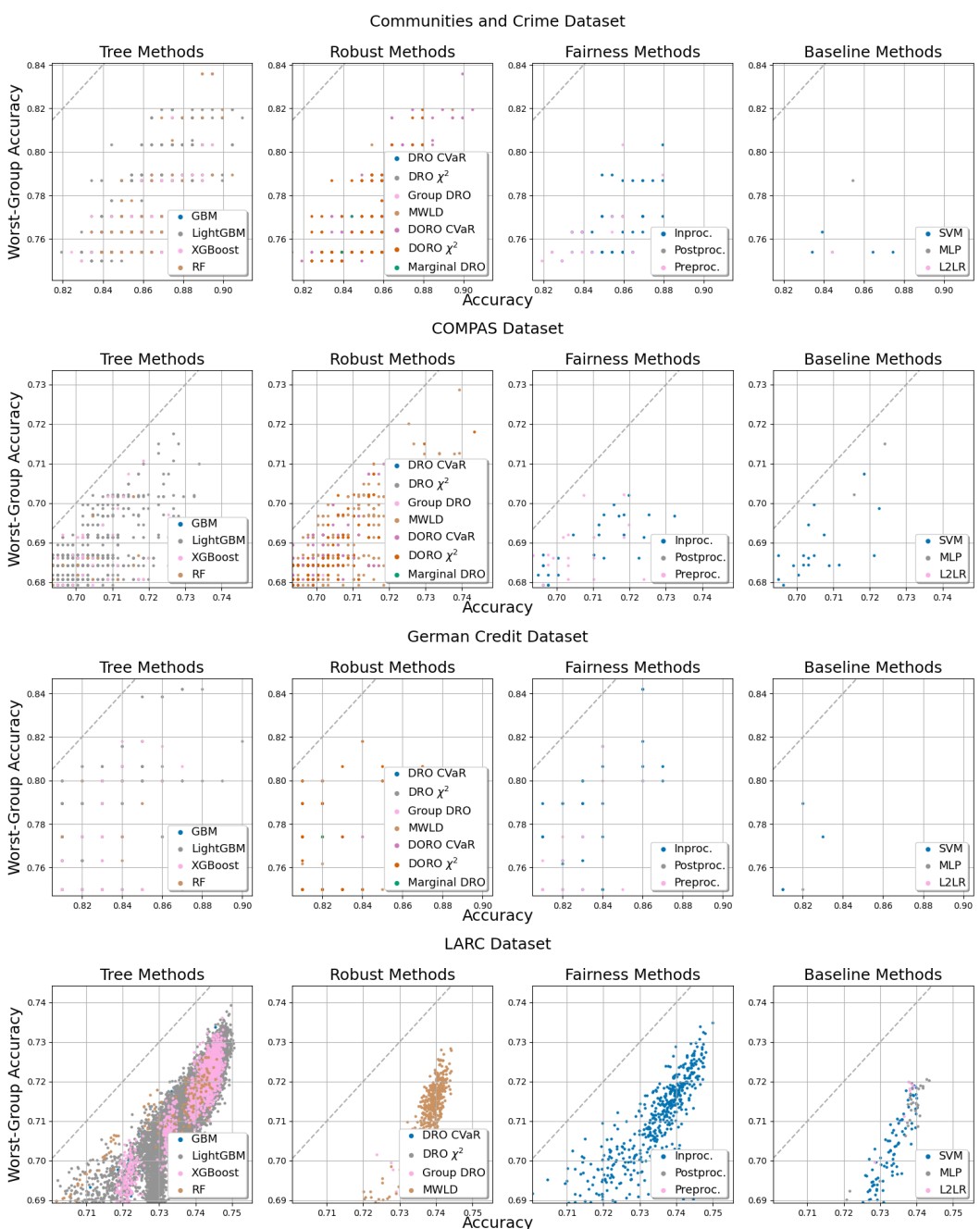

Figure 11: Overall Accuracy vs. Worst-Group Accuracy of robust, fairness-enhancing, tree-based, and baseline models over eight tabular datasets (Communities and Crime, COMPAS, German Credit, LARC). For results by individual algorithm, see Figures 12- 15.

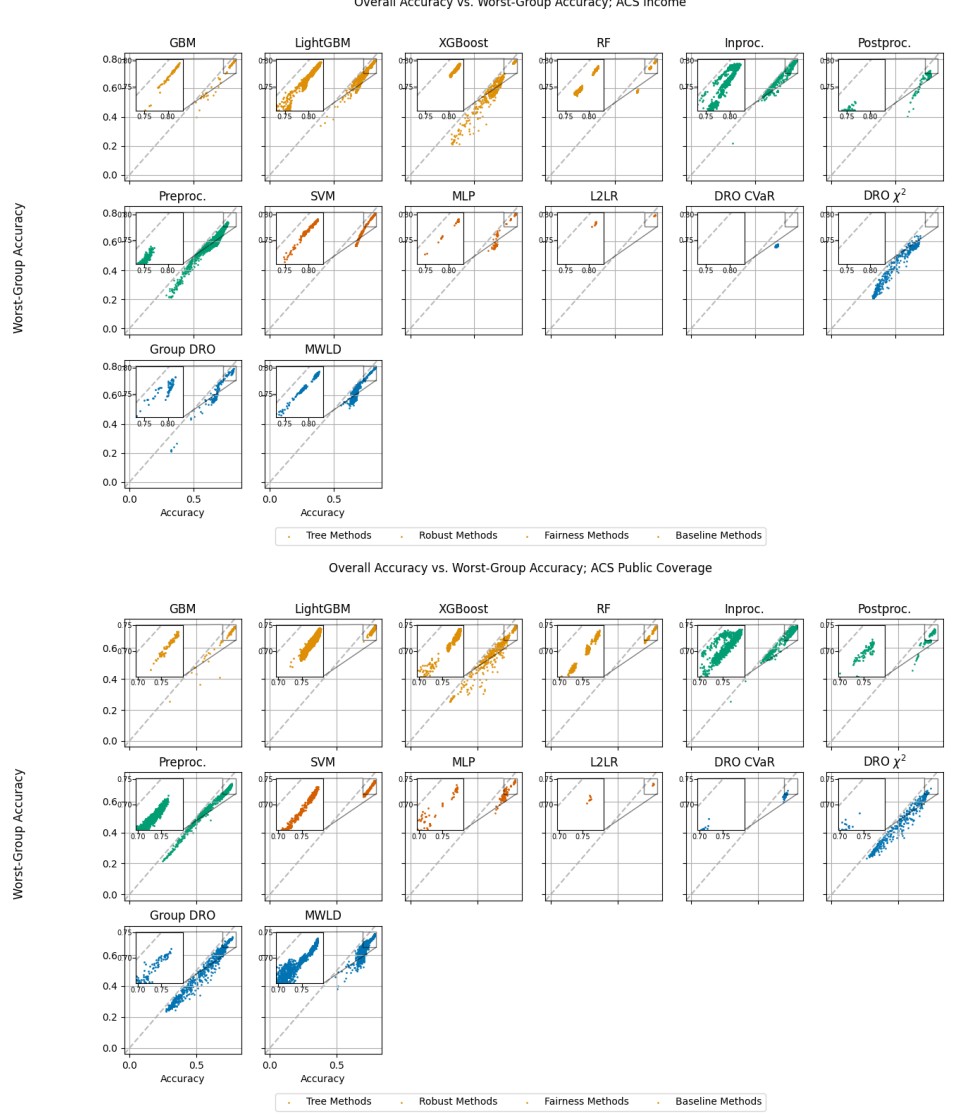

Figure 12: Detailed accuracy vs. worst-group accuracy for ACS Income and ACS Public Coverage datasets.

on the German Credit dataset, which we hypothesize is due to potential overloading of the CPU cluster during these training experiments (we note that German is, by far, the smallest dataset in our study with $n = 1000$, and the XGBoost algorithm generally performs well on small datasets).

Collectively, these results, combined with the demonstration that tree-based models also require fewer iterations to tune due to their decreased hyperparameter sensitivity (see Section 6), suggest that tree-based models are a considerably more resource-efficient way to achieve state-of-the-art subgroup robustness for tabular data classification.

## G.2 Peak Performance Summary

We provide the best performance per model, in terms of both overall accuracy and worst-group accuracy, in Tables 3 and 4, respectively.

|  | Income | Pub. Cov. | Adult | BRFSS | C&C | COMPAS | German | LARC |
|---|---|---|---|---|---|---|---|---|
| DORO CVaR | 0.816 | N/A | 0.852 | N/A | 0.905 | 0.725 | 0.86 | N/A |
| DORO $\chi^2$ | 0.813 | N/A | 0.851 | N/A | N/A | 0.743 | N/A | N/A |
| DRO CVaR | 0.677 | 0.719 | 0.778 | 0.898 | 0.879 | 0.627 | 0.83 | 0.644 |
| DRO $\chi^2$ | 0.694 | 0.742 | 0.774 | 0.896 | 0.869 | 0.691 | 0.82 | 0.667 |
| GBM | 0.824 | 0.786 | 0.873 | 0.896 | 0.854 | 0.712 | 0.82 | 0.747 |
| Group DRO | 0.816 | 0.77 | 0.85 | 0.897 | 0.894 | 0.702 | 0.82 | 0.729 |
| Inprocessing + GBM | 0.823 | 0.789 | 0.87 | 0.898 | 0.879 | 0.732 | 0.87 | 0.75 |
| L2LR | 0.815 | 0.768 | 0.852 | 0.895 | 0.844 | 0.7 | 0.82 | 0.739 |
| LightGBM | 0.827 | 0.791 | 0.874 | 0.901 | 0.91 | 0.734 | 0.9 | 0.751 |
| MLP | 0.821 | 0.782 | 0.857 | 0.897 | 0.879 | 0.724 | 0.82 | 0.743 |
| MWLD | 0.823 | 0.784 | 0.864 | 0.898 | 0.894 | 0.739 | 0.85 | 0.744 |
| Marginal DRO | N/A | N/A | N/A | N/A | 0.849 | 0.72 | 0.82 | N/A |
| Postprocessing + GBM | 0.775 | 0.772 | 0.842 | 0.897 | 0.749 | 0.689 | 0.85 | 0.714 |
| Preprocesing + GBM | 0.771 | 0.765 | 0.827 | 0.897 | 0.879 | 0.724 | 0.86 | 0.678 |
| Random forest | 0.82 | 0.786 | 0.865 | 0.897 | 0.905 | 0.728 | 0.85 | 0.746 |
| SVM | 0.821 | 0.785 | 0.857 | 0.898 | 0.874 | 0.723 | 0.83 | 0.74 |
| XGBoost | 0.824 | 0.79 | 0.875 | 0.899 | 0.894 | 0.725 | 0.88 | 0.748 |

Table 3: Best observed overall accuracy per model, by dataset.

|  | Income | Pub. Cov. | Adult | BRFSS | C&C | COMPAS | German | LARC |
|---|---|---|---|---|---|---|---|---|
| DORO CVaR | 0.788 | N/A | 0.81 | N/A | 0.836 | 0.71 | 0.8 | N/A |
| DORO $\chi^2$ | 0.783 | N/A | 0.808 | N/A | N/A | 0.718 | N/A | N/A |
| DRO CVaR | 0.582 | 0.672 | 0.712 | 0.857 | 0.803 | 0.606 | 0.762 | 0.492 |
| DRO $\chi^2$ | 0.641 | 0.683 | 0.707 | 0.863 | 0.789 | 0.676 | 0.789 | 0.61 |
| GBM | 0.796 | 0.738 | 0.833 | 0.851 | 0.787 | 0.684 | 0.737 | 0.734 |
| Group DRO | 0.783 | 0.718 | 0.807 | 0.855 | 0.816 | 0.682 | 0.789 | 0.702 |
| Inprocessing + GBM | 0.796 | 0.742 | 0.845 | 0.859 | 0.803 | 0.702 | 0.842 | 0.735 |
| L2LR | 0.785 | 0.718 | 0.808 | 0.849 | 0.754 | 0.644 | 0.727 | 0.72 |
| LightGBM | 0.798 | 0.745 | 0.837 | 0.87 | 0.836 | 0.718 | 0.842 | 0.739 |
| MLP | 0.791 | 0.738 | 0.814 | 0.854 | 0.787 | 0.715 | 0.75 | 0.721 |
| MWLD | 0.794 | 0.739 | 0.826 | 0.859 | 0.82 | 0.729 | 0.774 | 0.728 |
| Marginal DRO | N/A | N/A | N/A | N/A | 0.77 | 0.707 | 0.774 | N/A |
| Postprocessing + GBM | 0.72 | 0.723 | 0.809 | 0.857 | 0.639 | 0.623 | 0.75 | 0.632 |
| Preprocesing + GBM | 0.737 | 0.717 | 0.781 | 0.861 | 0.803 | 0.702 | 0.816 | 0.647 |
| Random forest | 0.79 | 0.741 | 0.824 | 0.858 | 0.836 | 0.702 | 0.8 | 0.726 |
| SVM | 0.791 | 0.735 | 0.815 | 0.858 | 0.763 | 0.707 | 0.789 | 0.719 |
| XGBoost | 0.796 | 0.744 | 0.836 | 0.868 | 0.836 | 0.711 | 0.818 | 0.736 |

Table 4: Best observed worst-group accuracy per model, by dataset.

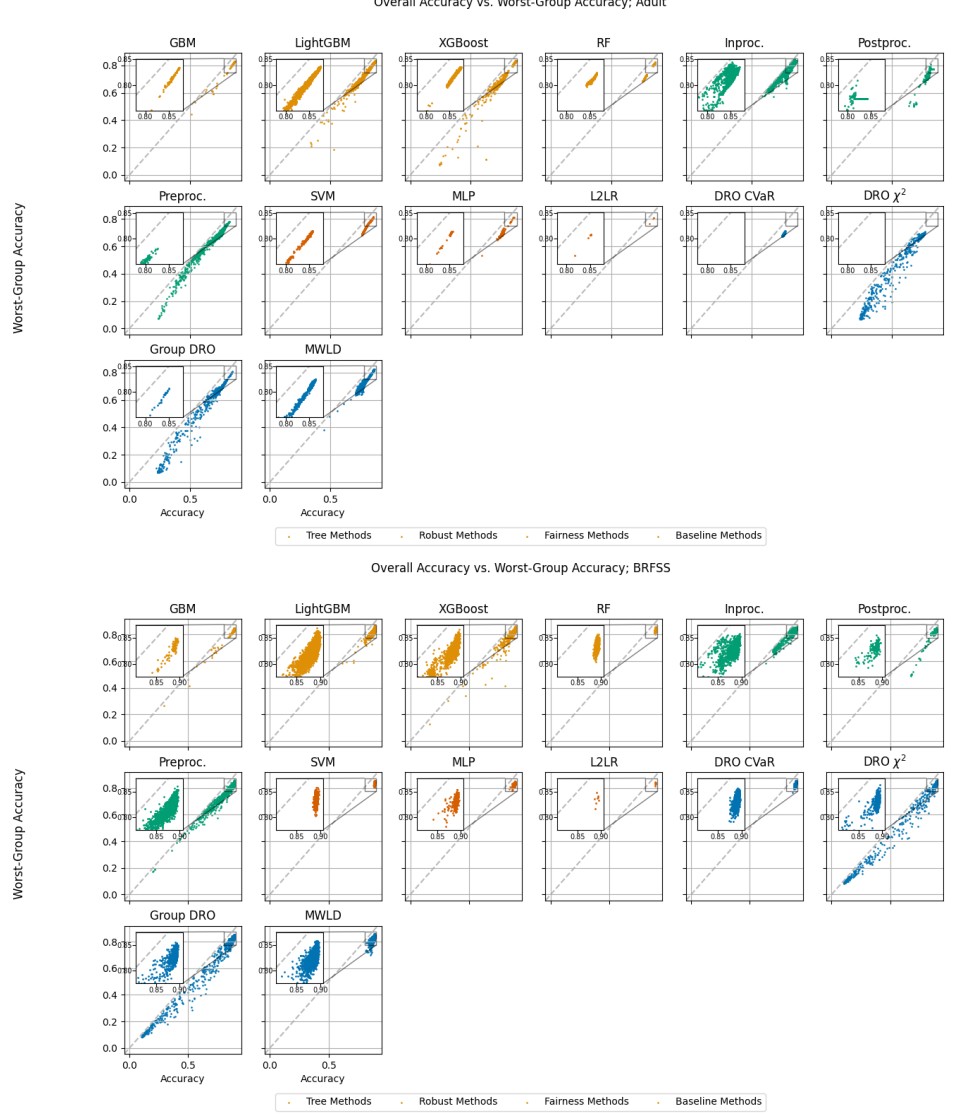

Figure 13: Detailed accuracy vs. worst-group accuracy for Adult and BRFSS datasets.

## G.3 Performance of Default Tree Hyperparameters

For the tree-based models in our summary, we give the performance of the default hyperparameters in Table 5.

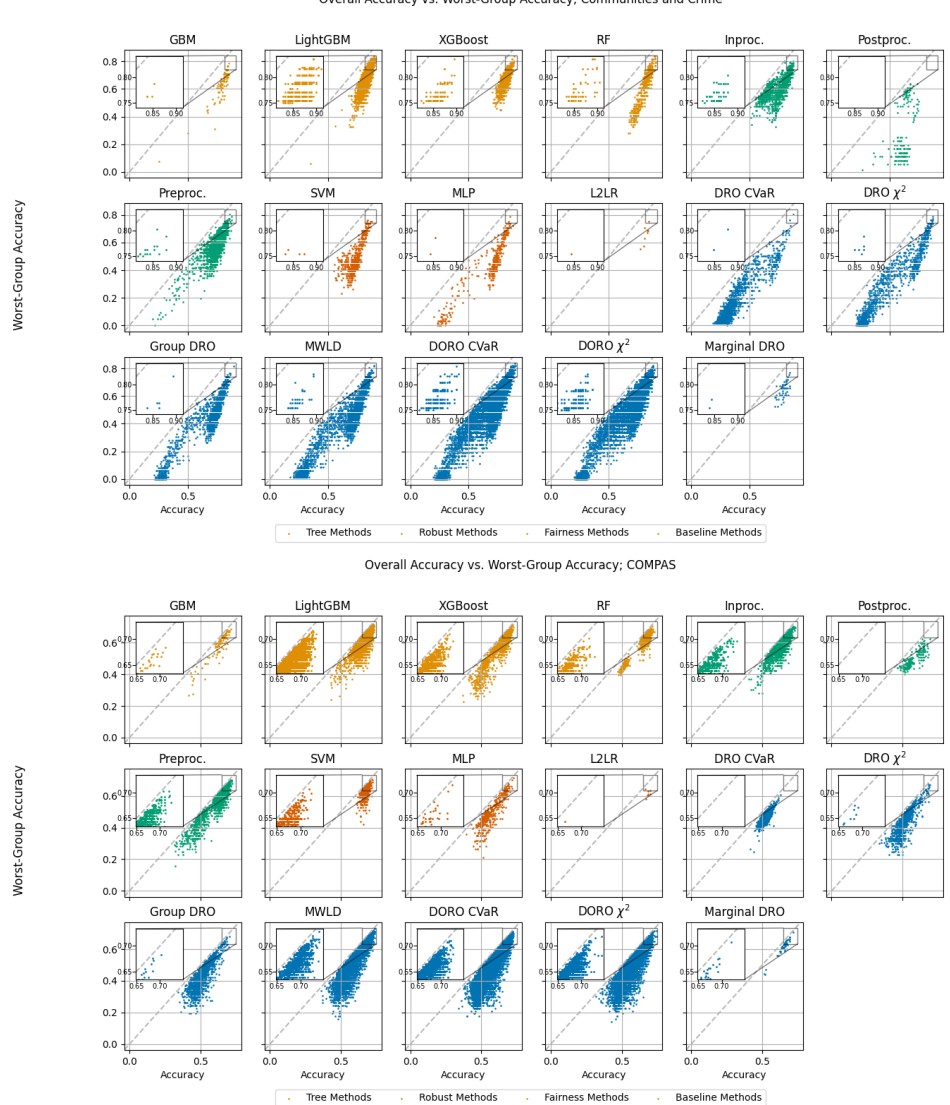

Figure 14: Detailed accuracy vs. worst-group accuracy for Communities and Crime and COMPAS datasets.

|  | Income | Pub. Cov. | Adult | BRFSS | C&C | COMPAS | German | LARC |
|---|---|---|---|---|---|---|---|---|
| **Overall Accuracy** | | | | | | | | |
| GBM | 0.81 | 0.777 | 0.871 | 0.892 | 0.859 | 0.7 | 0.79 | 0.739 |
| LightGBM | 0.823 | 0.787 | 0.874 | 0.887 | 0.819 | 0.682 | 0.75 | 0.747 |
| Random Forest | 0.812 | 0.765 | 0.852 | 0.891 | 0.854 | 0.669 | 0.74 | 0.754 |
| XGBoost | 0.825 | 0.788 | 0.872 | 0.893 | 0.854 | 0.698 | 0.73 | 0.748 |
| **Worst-Group Accuracy** | | | | | | | | |
| GBM | 0.779 | 0.725 | 0.83 | 0.86 | 0.711 | 0.684 | 0.65 | 0.718 |
| LightGBM | 0.793 | 0.738 | 0.834 | 0.834 | 0.658 | 0.667 | 0.6 | 0.725 |
| Random Forest | 0.781 | 0.713 | 0.808 | 0.848 | 0.738 | 0.596 | 0.71 | 0.742 |
| XGBoost | 0.797 | 0.735 | 0.834 | 0.833 | 0.754 | 0.654 | 0.55 | 0.728 |

Table 5: Performance of default hyperparameters for tree-based models.

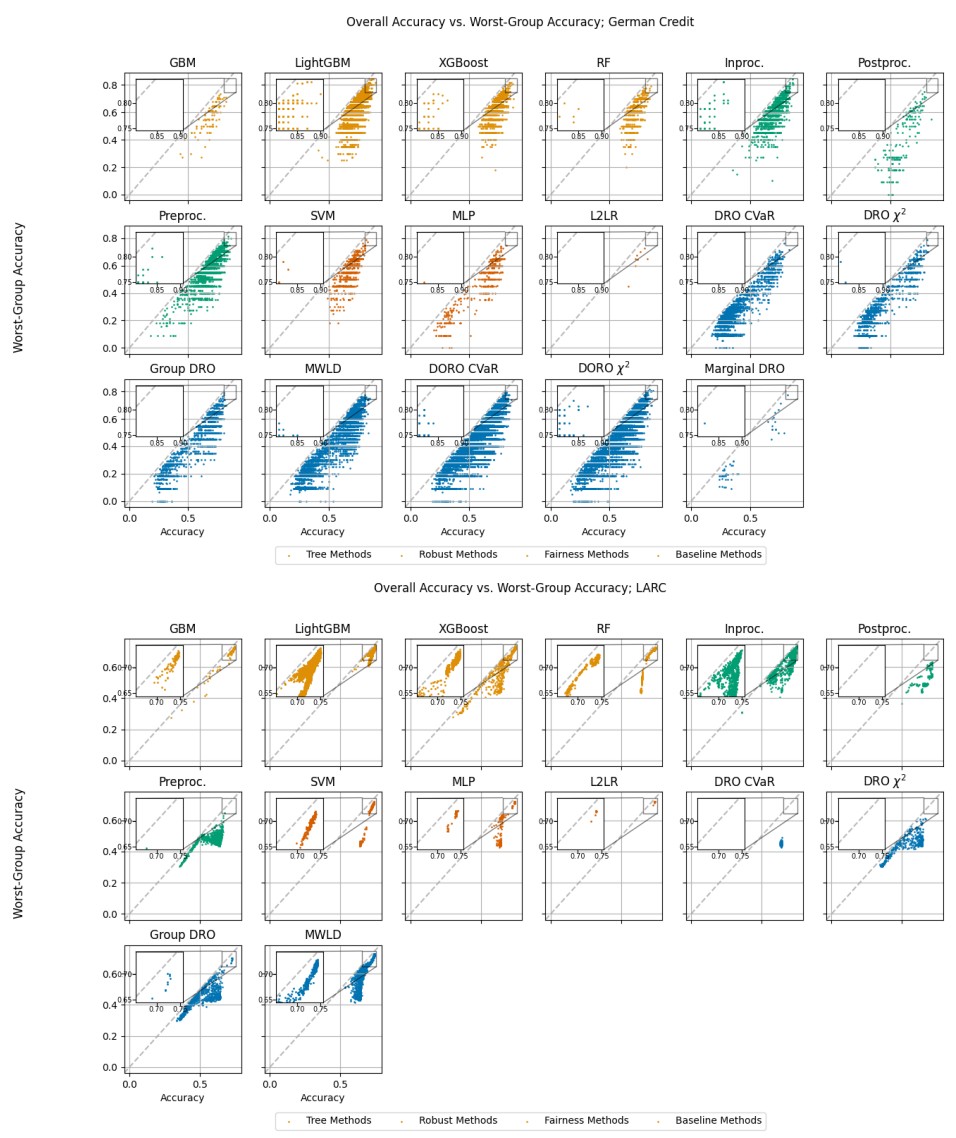

Figure 15: Detailed accuracy vs. worst-group accuracy for German and LARC datasets.

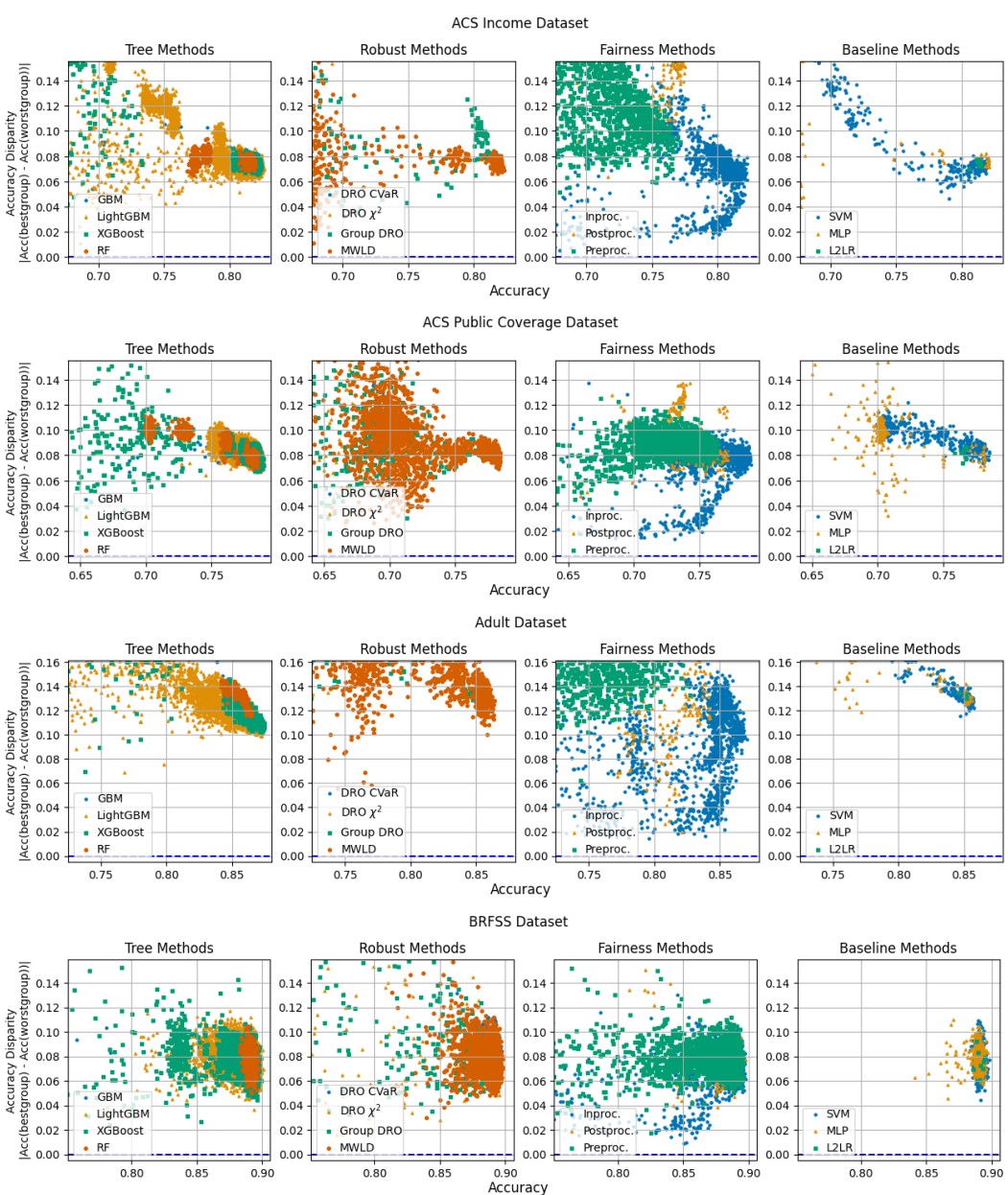

Figure 16: Overall Accuracy vs. Accuracy Disparity of robust, fairness-enhancing, tree-based, and baseline models over datasets ACS Income, ACS Public Coverage, Adult, BRFSS. See also Figure 17.

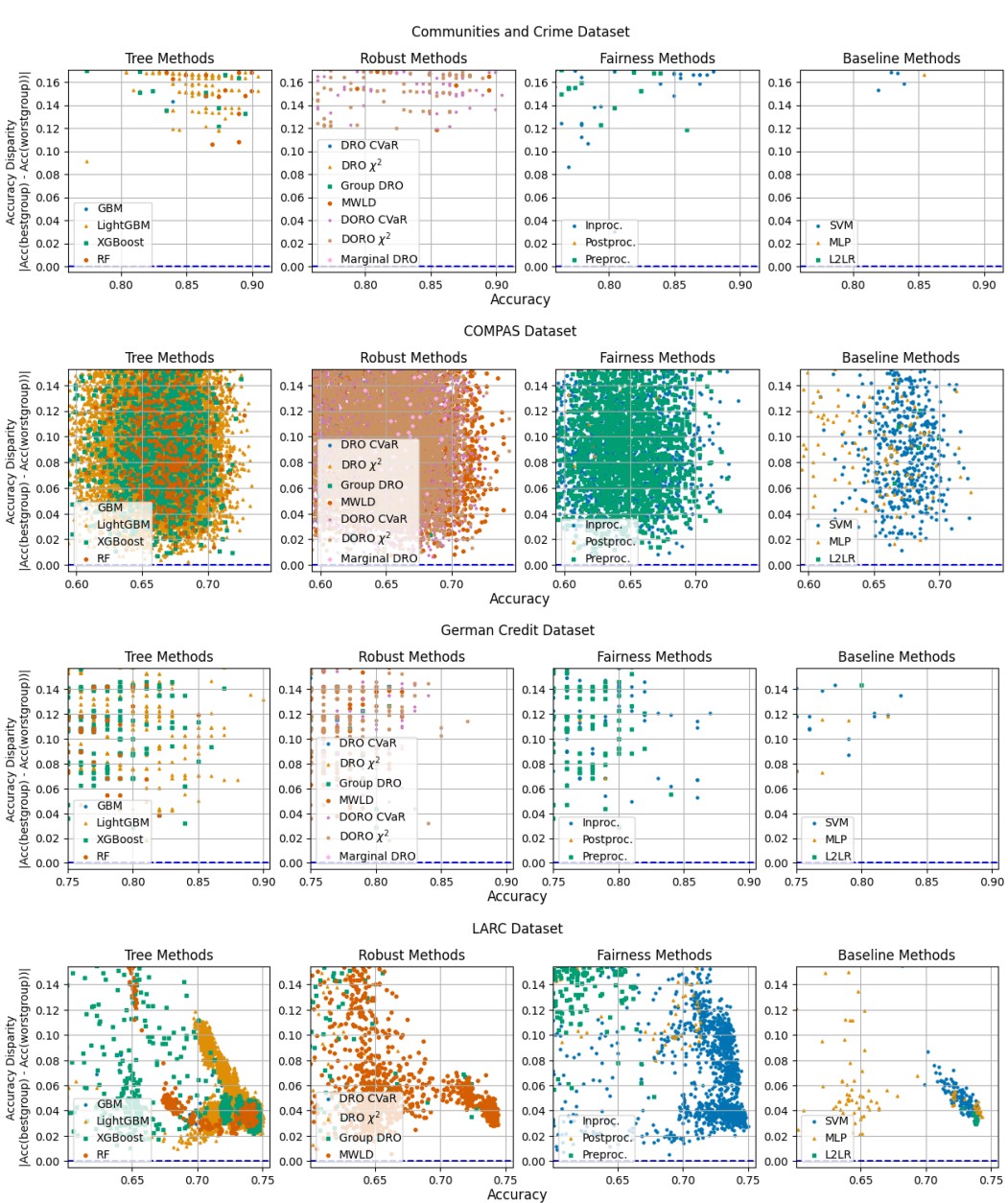

Figure 17: Overall Accuracy vs. Accuracy Disparity of robust, fairness-enhancing, tree-based, and baseline models over datasets Communities and Crime, COMPAS, German, LARC. See also Figure 16.

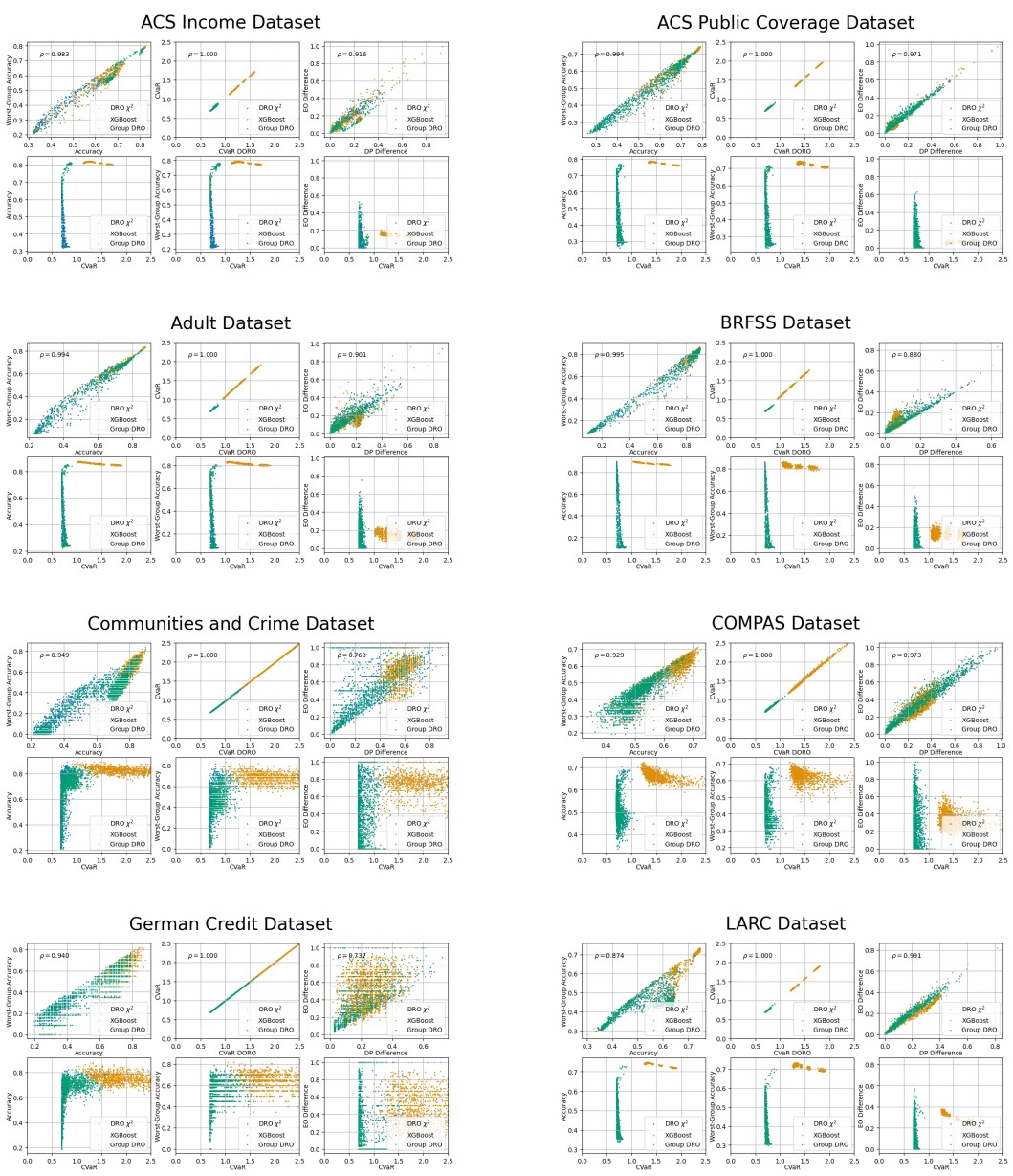

Figure 18: Correlation between complementary metrics (top row) and non-complementary metrics (bottom row) for each dataset, for DORO, XGBoost, and Group DRO models. Complementary metrics show strong correlations for all models, while non-complementary metrics do not. Pearson's $r$ correlation coefficient for each pair of complementary metrics shown in the upper-left of each plot.

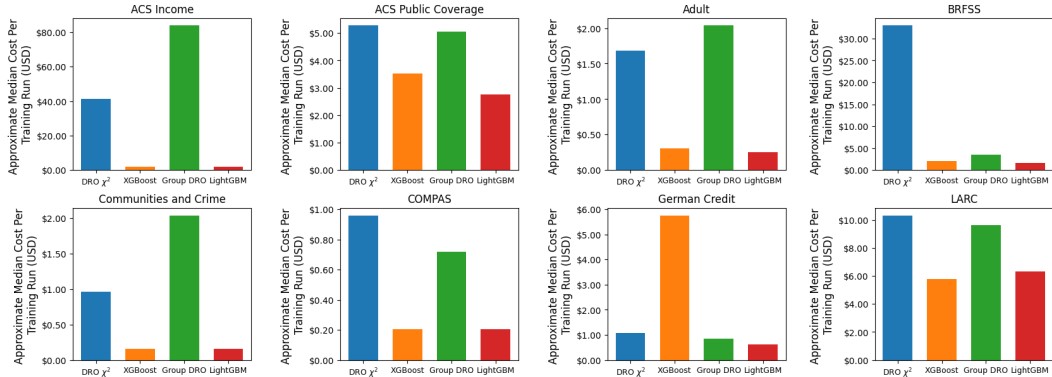

Figure 19: Estimated cost per training run, based on the median train time over the iterations in our study and the price of cloud-based computing infrastructure.