# OpenReview forum: "Subgroup Robustness Grows On Trees: An Empirical Baseline Investigation"
_NeurIPS.cc/2022/Conference — NeurIPS 2022 Accept_

### Official Review · Reviewer_Uv7V · 2022-07-11

**Rating:** 6
**Confidence:** 4
**Soundness:** 3 good
**Presentation:** 3 good
**Contribution:** 2 fair

**Summary:**

The paper empirically evaluates subgroup robustness methodologies on tabular data with clearly defined subgroups. They posit that state of the art tree-based methods are already robust learners in the subgroup-robust sense, and that they outperform the existing state of the art fairness and robustness methods in these categories. Further, they show that existing subgroup robustness methods can be brittle (i.e., show poor performance under other robustness criteria)

**Questions:**

What do the authors see as the main limitation for the standard robustness- and fairness-based methods to work on tree-based architectures?

Do the authors believe that tree-based methods cannot be further improved by the use of robust learning?

**Limitations:**

No dedicated section was provided, the work itself highlights potential areas for improvement in the area, which by itself is a net positive impact.

**Strengths And Weaknesses:**

Strengths:

The paper is clear in its goals and methodology, and the experiments show that worst group loss and worst disparity between subgroups are very well addressed by existing tree-based methods on tabular data, even outperforming many existing subgroup fairness and robustness approaches.

Weaknesses:

The vast majority of subgroup robustness and fairness algorithms are evaluated on MLP-based architectures. Though this is a limitation of current fairness and robustness work, it is hard to draw conclusions on the field at large other than the need for a greater focus on methods that work on tree-based architectures, or on better base architectures for tabular data specifically.

It is not clear to me whether these results are surprising, since a sufficiently large improvement in overall model accuracy is warranted to also improve worst-group accuracy.

---

> ### Author Response · Authors · 2022-08-02
> **Author Response to Reviewer Uv7V**
>
> We thank the reviewer for their comments. We are glad to see that the reviewer found our paper “clear in its goals and methodology” and that “the work itself highlights potential areas for improvement in the area, which by itself is a net positive impact.”
>
> We address reviewer comments and questions below.
>
> **MLP-based architectures:** The reviewer notes that our findings using neural network models are specific to MLP-based architectures. We agree. We offer the following additional comments:
>
> 1. Our goal was to fairly evaluate the models evaluated in prior robustness-oriented work. As the reviewer notes (“this is a limitation of current fairness and robustness work”), the only architectures used in the tabular-data experiments in these previous works (DRO, Group DRO, DORO) are MLPs. We thus saw this as a necessary component of conducting a fair evaluation of the previous works. We followed the reviewer’s suggestion to advocate for “a greater focus on methods that work on tree-based architectures, or on better base architectures for tabular data specifically” in our revisions.
> 2. We note that the suggestion that alternative architectures may be a promising approach to improving robustness is a result of our study, and thus a demonstration of its impact. The previous works mentioned above largely suggest that MLP-based architectures were effective subgroup-robust models; only the analysis in our study demonstrates that there are in fact better alternatives.
> 3. We believe that a thorough investigation of alternative (non-MLP) deep learning architectures, in combination with robust optimization methods, is beyond the scope of the current work. Further, as the works surveyed in Section 2.3 demonstrate (e.g. [1], [2]), non-MLP-based architectures have failed to outperform MLP-based models in large-scale empirical studies, while MLPs have been shown to be capable of SOTA performance for tabular data [3].
>
> **Surprisingness of findings:**  We recognize that the reviewer is unsure whether this result is “surprising.” We offer the following responses:
> * Our demonstration that well-known, widely-used tabular data models outperform the existing SOTA for robust learning (i.e. DORO) is quite surprising; it was certainly unexpected for us given the considerable research effort dedicated to developing existing robust learning techniques. For example, [4] reports improved average and worst-group accuracy using DORO (cf [4], Fig 2(i), 2(j)], and [5] reports improvements in overall and worst-group accuracy from the use of Group DRO (cf. [5], Table 1).
> * Before our study, we agree that there was not a clear strong prior indication as to whether trees would show comparable subgroup robustness to SOTA robust learning methods. However, this literature gap is precisely what motivates the need for our work.
>
> **Questions:**
>
> * *"What do the authors see as the main limitation for the standard robustness- and fairness-based methods to work on tree-based architectures?"* Tree-based models are nondifferentiable, and are thus incompatible with fully gradient-based training required to use robustness-enhancing methods.
> * *"Do the authors believe that tree-based methods cannot be further improved by the use of robust learning?"* We do not have a strong intuition here. Our results show that in many cases trees already approach the Pareto frontier of zero subgroup disparity (e.g. Fig. 1b; direct link below). However, we believe it is worthy of further investigation, and that this question can be answered empirically in future work, as we note in Section 7.
> * *Limitations:* We now include a discussion of limitations in the main text in Section 7. Please also see our more detailed discussion of limitations in the shared response above.
>
> ### Citations:
>
> [1] Shwartz-Ziv, Ravid, and Amitai Armon. "Tabular data: Deep learning is not all you need." Information Fusion 81 (2022): 84-90.
>
> [2] Borisov, Vadim, et al. "Deep neural networks and tabular data: A survey." arXiv preprint arXiv:2110.01889 (2021).
>
> [3] Kadra et al. Well-tuned Simple Nets Excel on Tabular Datasets. NeurIPS 2021. https://arxiv.org/abs/2106.11189
>
> [4] Zhai, Runtian, et al. "Doro: Distributional and outlier robust optimization." International Conference on Machine Learning. PMLR, 2021.
>
> [5] Sagawa, Shiori, et al. "Distributionally robust neural networks for group shifts: On the importance of regularization for worst-case generalization." arXiv preprint arXiv:1911.08731 (2019).
>
> ### Direct links to updated figures from main text:
>
> * Figure 1: https://i.postimg.cc/fR4L8cS1/Fig1.png (main results summary)

---

### Official Review · Reviewer_WCHM · 2022-07-11

**Rating:** 5
**Confidence:** 3
**Soundness:** 3 good
**Presentation:** 4 excellent
**Contribution:** 2 fair

**Summary:**

Through extensive experiments, the authors demonstrate that tree-based methods are a strong and computationally frugal baseline for tabular prediction, ultimately achieving favorable performance in terms of overall and subgroup performance.


**Questions:**

Race and gender and gender are listed as examples of binary sensitive attributes [line 140]. While these variables may be codified as binary in the dataset, they are not binary in reality [https://arxiv.org/abs/2205.04610, https://dl.acm.org/doi/10.1145/3411764.3445742]. Making this distinction is important, especially in settings where the data collection is beyond our control (e.g. fixed benchmark datasets).
Typo in line 273: “evaluated basedon”


**Limitations:**

I think the authors could do more to contextualize the specific tabular prediction tasks evaluated. For example the use of tabular methods for risk assessment and neighborhood crime-level prediction could have negative societal impacts, even if they achieve some notion of statistical fairness or subgroup robustness. Cf. https://arxiv.org/abs/2106.05498


**Strengths And Weaknesses:**

Strengths
* The authors investigate subgroup robustness properties of tree-based models on tabular prediction, which I haven’t seen before and believe will be interesting to the neurips community (although tree-based methods are known to be a strong baseline for tabular benchmarks in general).
* Quality of presentation is high. Baseline methods and empirical results are presented in an accessible way.
* Very nice related works section that would serve as a useful mini-survey for the interested reader.

Weaknesses
* What seems to be missing is some reasonable hypothesis for *why* tree-based methods tend to find subgroup-robust models. Cynthia Rudin has an extensive body of work arguing that tree-based methods (e.g. https://www.ncbi.nlm.nih.gov/pmc/articles/PMC9122117/)  are very effective modeling tabular data, and may be better equipped to find a good solution than deep learning based approaches. The results presented here are consistent with that hypothesis since we see that the tree-based methods are best at robust metrics precisely because they are best at overall accuracy, and the authors show that aggregate accuracy and subgroup accuracy are very correlated on this data. It makes me think that the main takeaway could be that trees are effective tabular learners, rather than subgroup-robust learners (although these are not mutually exclusive).

---

> ### Author Response · Authors · 2022-08-02
> **Author Response to Reviewer WCHM**
>
> We are grateful to the reviewer for their useful comments. We are pleased that the reviewer states that they haven’t seen such an analysis before, that it “will be interesting to the neurips community,” noting that “[q]uality of presentation is high” and the presentation of results “accessible” with a “[v]ery nice related works section that would serve as a useful mini-survey for the interested reader.”
>
> The reviewers’ primary concern with the paper is a lack of a clear explanation “for why tree-based methods tend to find subgroup-robust models”.
>
> In our shared response, above, we highlight some potential explanations, which we have also added to the main text. Broadly, we agree with the reviewers’ suggestion, also supported by prior work (those cited in our paper, and the provided Rudin references), that the observed subgroup robustness is due, in part, to simply fitting the data better. We have added the references suggested by the reviewer to our work.
>
> However, we note that none of the cited works -- in our paper or provided by the reviewer -- investigate this claim with respect to subgroup robustness. This is particularly important, we feel, given the recent developments of specialized methods designed to imbue supervised learning models with subgroup robustness (i.e. Group DRO, DORO, etc.), along with a prevailing notion that “fairness” or subgroup robustness and accuracy may actually trade off, instead of correlate [1, 2, 3, 4]. We also note that prior work on robustness exists based on the assumption that just "fitting the data" is not enough (if it was enough, there is no need for robustness methods, and we could strictly use ERM), and this often backed by empirical evidence [e.g. 5].
>
> We address additional reviewer comments below:
>
> * *Questions:* we certainly agree with the reviewer regarding the reduction of these non-binary sensitive attributes. We added a statement to this effect, including the suggested references, to the section “Dataset Details”, with a more explicit statement/reference to these important considerations in main text Section 3.4.
>
> * *Limitations:* This is related to the above; we are grateful to the reviewer for raising this important point and have added to the same sections of text noted above. In particular, we added the Bao et al. reference to our existing discussion of concerns about COMPAS in particular (see Section A, “COMPAS”). We mostly use COMPAS as a key point of comparison to previous works which use this dataset, despite the flaws in the dataset itself.
> * *Takeaways:* The reviewer notes that “​​the main takeaway could be that trees are effective tabular learners” -- we agree that this is *a* takeaway of our work, as our study provides clear evidence to support this conclusion, but we believe that the *main* takeaway is that trees are subgroup-robust learners and match or outperform existing SOTA robust learning methods, since the tabular classification performance of trees is well-established in prior work (see Sec. 2.3).
>
> ### References
>
> [1] Friedler, Sorelle A., et al. "A comparative study of fairness-enhancing interventions in machine learning." Proceedings of the conference on fairness, accountability, and transparency. 2019.
>
> [2]  Menon, Aditya Krishna, and Robert C. Williamson. "The cost of fairness in binary classification." Conference on Fairness, Accountability and Transparency. PMLR, 2018.
>
> [3] Chen, Irene, Fredrik D. Johansson, and David Sontag. "Why is my classifier discriminatory?." Advances in neural information processing systems 31 (2018).
>
> [4] Zhao, Han, and Geoff Gordon. "Inherent tradeoffs in learning fair representations." Advances in neural information processing systems 32 (2019).
>
> [5] Zhai, Runtian, et al. "Doro: Distributional and outlier robust optimization." International Conference on Machine Learning. PMLR, 2021.

---

> > ### Comment · Reviewer_WCHM · 2022-08-05
> > **quick question**
> >
> > Thanks to the authors for their considered response, and for the updates to the paper.
> >
> > Regarding the assumption of prior work that just fitting the data is not enough: do you see your paper as inheriting this assumption, or repudiating it?

---

> > > ### Author Response · Authors · 2022-08-06
> > > **Author response to reviewer WCHM question**
> > >
> > > We thank the reviewer for the interesting question.
> > >
> > > First, we would like to note that it may be impossible to give a binary answer to a broad hypothesis such as “fitting the data is not enough.” Whether just fitting the data is enough may depend crucially on the data in question, so the assumption may be true on some datasets and wrong on others. Understanding when, not if, the assumption holds is then the main scientific question. Since we currently lack a systematic categorization of tabular data, the first step to answer this question is comprehensive evaluations on many datasets and distribution shifts. Our experiments hope to take a step in this direction.
> > >
> > > With this in mind, our work was certainly initially motivated by the assumption that principled subgroup robustness methods could do better than methods without an explicit subgroup robustness objective. We soon discovered that the tree baselines achieved excellent performance in comparison to the robustness methods. Additionally, the assumption that fitting the data is not enough is fundamental to the prior robustness literature: as we mention, if "fitting the data" was enough, there would be no need for robustness-based interventions.
> > >
> > > While our work does not seek to repudiate the assumption that fitting the data is not enough, we do believe our results call into question whether existing robustness algorithms are achieving improved subgroup robustness on the subgroups that exist in real-world data. In our experiments, on no dataset does an existing robustness intervention outperform XGBoost (cf. Figure 3), despite the lack of any explicit robustness-enhancing elements in the XGBoost algorithm. This at least calls the assumption into question with the currently-available set of robust learning methods.
> > >
> > > Our work does not eliminate the possibility that we can do more than "fit the data." It is quite possible that better subgroup robustness methods exist: even XGBoost does not achieve zero disparity in many cases (as demonstrated by the 95% CIs for the XGBoost curve in Figure 3 not containing the “zero disparity line” in multiple datasets, particularly for the highest-accuracy points on the curves). We think there is still space for future work to investigate this assumption. We believe that evidence prompts us to rethink this assumption, and providing guidance on future work to investigate it (e.g. by focusing on tree-based models or those with similar properties; moving away from MLP-only architectures), is a contribution of our benchmarking study.
> > >
> > > * Direct Link to Figure 3: https://i.postimg.cc/bwQvWZQr/Fig3.png

---

> > > > ### Comment · Reviewer_WCHM · 2022-08-09
> > > > **on data fitting**
> > > >
> > > > Thanks for responding; I wasn't trying to stump you with a "gotcha" question, and I appreciate the nuance of discussion in your answer. My current thinking, based on reading your paper, is that tabular settings are very difficult, which motivates fitting the data first as a proxy for subgroup performance, which in turns motivates the choice of model class known to fit tabular data well (trees, forests, etc.).
> > > >
> > > > For what it's worth, the assumption that "fitting the data is not enough" has not gone unchallenged in the domain generalization literature (even for vision tasks). For example Gulrajani and Lopez-Paz 2021 showed that ERM is a very difficult baseline to beat (even by domain-aware methods like IRM and CORAL) when model selection is done carefully and in a way that is fair to all methods.
> > > >
> > > > I don't think it's reasonable for me to ask for this in a rebuttal, but for a future version of this paper I'm curious about how tree-based methods would handle more drastic distribution shifts for example in the IRM paper they propose a semi-synthetic dataset (CMNIST) that is essentially designed for ERM to fail. We can certainly scrutinize that dataset, but it at least highlights a failure mode of ERM. And I wonder if a carefully constructed tabular task could highlight failure modes of tree-based methods to tell us whether domain/subgroup-aware methods *can* be useful in extreme settings.
> > > >
> > > > References:
> > > > [Gulrajani and Lopez Paz 2021] *In Search of Lost Domain Generalization*, ICLR 2021, https://openreview.net/forum?id=lQdXeXDoWtI

---

> > > > > ### Author Response · Authors · 2022-08-09
> > > > > **Author response to "on data fitting"**
> > > > >
> > > > > We thank the reviewer for the interesting dialogue. Providing a strong empirical foundation for formulating and testing hypotheses in distribution shift is a key goal of our work, and we are glad that the work has stimulated this dialogue.
> > > > >
> > > > > We agree that “In Search of Lost Domain Generalization” contains a very interesting set of results. Other work on distribution shift in image classification has also pointed out that fitting the data is at least sometimes enough, e.g., see [1, 2]. We note however, that both “In Search of Lost Domain Generalization” and [1, 2] only deal with computer vision tasks, while our focus is on tabular data. Hence the experiments in our work differ from prior work in important ways (type of datasets, methods, etc.), and it is not clear a priori that findings translate between image data and tabular data (e.g., methods often do *not* transfer between the two domains). We will clarify this in the updated version of our paper and make sure to cite the related work in computer vision.
> > > > >
> > > > > We also agree that “testing the limits” of subgroup robustness would be an interesting next step, and that datasets with intentional, systematic shifts would be useful in understanding the limits of the subgroup robustness of tree-based methods. Our focus here has been on shifts occurring naturally in real datasets as these shifts are usually the most relevant from an application perspective. Nevertheless, we are grateful for the suggestion and hope to investigate this question in future work because additional experiments of this scale are outside the scope of our paper (as the reviewer notes).
> > > > >
> > > > >
> > > > > References:
> > > > >
> > > > > [1] Taori, Rohan, et al. "Measuring robustness to natural distribution shifts in image classification." Advances in Neural Information Processing Systems 33 (2020): 18583-18599.
> > > > >
> > > > > [2] Miller, John P., et al. "Accuracy on the line: on the strong correlation between out-of-distribution and in-distribution generalization." International Conference on Machine Learning. PMLR, 2021.

---

### Official Review · Reviewer_kdGJ · 2022-07-12

**Rating:** 5
**Confidence:** 4
**Soundness:** 3 good
**Presentation:** 2 fair
**Contribution:** 2 fair

**Summary:**


This paper provides an extensive empirical analysis of the subgroup robustness of tree-based models (like XGBoost), distributionally robust models (like DRO), and fairness-promoting methods (like fairlearn) on five commonly used datasets for fairness-aware machine learning: ACS Income, Adult, Communities and Crime, COMPAS, and German Credit. For each model type, they train many models over a grid of hyperparameter values, and they report the worst-group accuracy, accuracy, max disparity between groups, and cVaR (a measure of accuracy in the tail). They find evidence that tree-based models show good subgroup-robustness properties, often on par with distributionally robust methods and with fairness-aware methods.

**Questions:**

- Can you provide any discussion on why grid search over tree hyper parameters seems to trace out the fairness-accuracy Pareto front?
- Similarly, can you provide motivation for why trees might have subgroup robustness?
- Is it possible to update the main text to include some details on the hyperparameter sweeps and other important experimental details?

**Limitations:**

There is no proper discussion of the limitations of this analysis. I would recommend that the authors add this. For instance, what are the limitations in extrapolating these findings to other datasets, other domains?

**Strengths And Weaknesses:**

# Originality

This empirical investigation is original and needed. Prior work (discussed in the paper) had conducted an empirical comparison of fairness methods early in the literature, but the modern update to include distributionally robust methods is original.

However, some of the claims of contributions in the paper are not well supported. The paper claims to “introduce techniques for the analysis of hyperparameter sweets such as model performance frontier curves” (line 62-63), but it is not clear how this is different from commonly reported Pareto frontier curves.

# Quality

Given recent works offering more extensive and more realistic datasets, I am not convinced that it is sufficient to perform this empirical analysis on the five traditional fairness datasets. A comprehensive empirical analysis of the Adult dataset should include analysis on other “folktables” as provided in Ding et al. Additionally, Koh et al. curated a list of datasets that contain “in-the-wild” distribution shifts including subpopulation shift. This seems like an important set of datasets to perform analysis on.

No uncertainty estimates are provided. In the checklist the authors claim they are “not concerned with point estimates” but most of the metrics reported here are point estimates, right? Relatedly, it may be helpful to present results across datasets—is there a way to graphically summarize the range of results you see across datasets? This might be helpful for readers who want to get a quick overall summary of the results.

Ding, Frances, et al. "Retiring adult: New datasets for fair machine learning." Advances in Neural Information Processing Systems 34 (2021): 6478-6490.

Koh, Pang Wei, et al. "Wilds: A benchmark of in-the-wild distribution shifts." International Conference on Machine Learning. PMLR, 2021.

# Clarity

The paper has a strong introduction with a nice discussion of the relationship between fairness and robustness.

However I generally found the remainder of the paper hard to read. For the exposition of the methods, it would be helpful to provide inline text describing the fairness and robust methods you include. It would also be useful to provide more discussion/intuition about why trees seem to have this robustness. It may be relevant to discuss prior work that analyzes when fairness-aware methods can address subpopulation shift:

Maity, Subha, et al. "Does enforcing fairness mitigate biases caused by subpopulation shift?." Advances in Neural Information Processing Systems 34 (2021): 25773-25784.

Singh, Harvineet, et al. "Fairness violations and mitigation under covariate shift." Proceedings of the 2021 ACM Conference on Fairness, Accountability, and Transparency. 2021.

 I found the designation of “complementary” metrics (line 245) to be confusing, and it would be helpful to add more details here to explicitly define what makes a metric complementary to another (and perhaps elaborate on an example).

I found it hard to digest many metrics reported over many graphs, and at times hard to make sense of the reported trends. For instance in Figure 1a the XGboost curve seems to track the y=x line, so does this indicate XGBoost had the same accuracy across all groups? Additionally, it was unclear to me what the difference between graph 1a and 1b was. It would be helpful to elaborate in the captions, including describing what the baselines refer to (or at least a pointer to the description in the text).


# Significance

This paper has the potential to be quite significant. Subgroup robustness is desired in many applications and their contribution showing that trees can have subgroup robustness is notable. The significance would be even stronger if the paper offered concrete takeaways/recommendations: e.g., in practice how should one make use of the findings here? For this it may be helpful to discuss the hyperparameter grid sweep in the main paper.

---

> ### Author Response · Authors · 2022-08-02
> **Author Response to Reviewer kdGJ, Part 1**
>
> We appreciate the reviewers’ thoughtful comments, and are pleased that the reviewer felt our work is “original and needed”, found our work’s potential contributions to be “quite significant” and findings “notable”. We respond to each comment below, but would also refer the reviewer to our overall response where we give more detailed discussions of (1) explanation of findings, (2) concrete takeaways/recommendations, and (3) limitations.
>
> **Contributions:** The reviewer raises concerns that our claims of contribution are “not well supported,” asking about the novelty of our hyperparameter analysis techniques. We agree that Pareto frontiers are not a novel technique; we have clarified the text helpfully identified by the reviewer. We reemphasize that our core contribution is our large-scale empirical comparison of several previously-disparate methods for supervised learning on tabular data and our demonstration that tree-based models are an effective subgroup-robust baseline.
>
> **Datasets:** The reviewer raises a valid concern about the use of five standard fairness datasets. We offer a few considerations here.
>
> 1. Being a benchmark study, it was critical to compare prior works on the datasets used in the original papers (i.e. DORO, DRO, Group DRO, etc.) to provide a direct comparison to those works and to well-established performance baselines on the datasets. We see these datasets as necessary, even if perhaps not sufficient, for a thorough empirical baseline.
> 2. Due to the large computational expense of running our experiments (training over 300k models independently), we needed to constrain our experiments to a feasible set of initial datasets. Each sweep consumes hundreds of GPU- and CPU-hours per dataset, so we focused our initial experiments on the datasets of greatest interest to the field.
> 3. We have added to our paper results from three additional datasets to our analysis, which include new domains (“LARC” - education, “BRFSS” - healthcare, “ACS Public Coverage” - policy). Our findings on these datasets confirm and indeed strengthen our original findings (see Figures 2, 3, 4, 6). As the reviewer helpfully suggested, we added the ACS task PublicCoverage from folktables [1], and we also changed the examples in Figure 1 and 5 from “Adult” dataset to instead focus on the folktables “ACS Income” task, which is their updated/improved version of the Adult dataset (and which has reduced uncertainty, due to considerably larger sample size). Direct links to these figures are below.
> 4. While we are aware of the useful benchmarking datasets of [2], none of the datasets in “Wilds” are tabular (which was one motivation for this work, as we feel this area is understudied relative to its importance). As such, we cannot include these in our study.
>
> **Uncertainty Estimates:** The quantification of the statistical uncertainty around our results is a critical point, and we are grateful to the reviewer for raising this concern, as addressing it makes our paper stronger. As noted in our main response, we have added principled statistical uncertainty estimates to all summary analyses in the main text (using Clopper-Pearson or Difference-in-Proportions CIs with $\alpha=0.05$). We note that this does not change the claims in our paper, but provides stronger evidence for them: for example, the CIs for XGBoost models either overlap or dominate those for DRO and Group DRO in the pareto analysis in Figures 1/3/4; we also now clearly see statistically-significant drops in performance for the non-tree models in the model selection results in Figure 6.
>
> **Takeaways:** We appreciate the reviewer’s helpful suggestion to clearly highlight takeaways and recommendations for practice. We discuss key takeaways from our study in the main response above, and note that these are now foregrounded in both the abstract and Conclusions section (7).
>
> **Intuition and “why” behind the results:** We agree with the reviewer that our study has the potential to do more to provide intuition or hypotheses to explain our observations. We discuss intuition and explanations for our observed findings in the main response above. We emphasize that the goal of our study is not to validate hypotheses about what might cause this improved robustness, but is to rigorously establish the state of the art in subgroup robustness and conduct an apples-to-apples evaluation of modern methods. We are particularly grateful for the citations suggested by the reviewer. In particular, the Maity et al. provides support for our hypothesis that tree-based ensembles improve subgroup robustness through improved overall performance, and that fairness constraints can harm subgroup robustness.

---

> > ### Author Response · Authors · 2022-08-02
> > **Author Response to Reviewer kdGJ, Part 2**
> >
> >
> > **Additional reviewer questions/comments:**
> > * “[I]n Figure 1a the XGboost curve seems to track the y=x line, so does this indicate XGBoost had the same accuracy across all groups?” Yes, the y=x line would be zero disparity. We added this to the legend, for this and all Figures of this form, to clarify.
> > * “Can you provide any discussion on why grid search over tree hyper parameters seems to trace out the fairness-accuracy Pareto front?” We would refer the reviewer to the shared response part (1), along with the previous bullet point, here. We note that our main contribution is to identify this behavior in a rigorous and controlled setting. The fact that this question is raised about tree models like XGBoost (and not, for example, DORO) we believe is an indication of our contribution.
> > * “[I]t was unclear to me what the difference between graph 1a and 1b was. It would be helpful to elaborate in the captions…” Figure 1(b) plots the pareto frontier of 1(a) to simplify the plots, for the three models. We have included this in the caption.
> > * “Is it possible to update the main text to include some details on the hyperparameter sweeps and other important experimental details?” We have added further details on the hyperparameter sweeps to the main text describing our procedure for widening continuous hyperparameter grids so that the best configurations are always in the interior of the grid, and we added Figures 7, 8 evaluating the hyperparameter sensitivity of the models in our study.
> > * “[I]t would be helpful to provide inline text describing the fairness and robust methods you include.” We acknowledge the reviewers’ point that more detail in the main text would be useful; given the space constraints, we favored extensive results presentation over detailed descriptions of preexisting and well-known baseline models. We will add more detail on the baseline methods in our camera-ready version.
> > We have added clarifications, additional annotations, and typo corrections to text, figures and their corresponding captions to address the reviewers’ concerns with clarity of the results. Additionally, we have updated our definition of “complementary metrics” in response to the reviewers’ concerns; we agree that the initial presentation could be improved.
> >
> >
> >
> > ### References:
> >
> > [1] Ding, Frances, et al. "Retiring adult: New datasets for fair machine learning." Advances in Neural Information Processing Systems 34 (2021): 6478-6490.
> >
> > [2] Koh, Pang Wei, et al. "Wilds: A benchmark of in-the-wild distribution shifts." International Conference on Machine Learning. PMLR, 2021.
> >
> >
> > ### Direct links to updated figures from main text:
> >
> > * Figure 1: https://i.postimg.cc/fR4L8cS1/Fig1.png (main results summary)
> > * Figure 2: https://i.postimg.cc/NMPGmmwJ/Fig2.png (overall vs. worst-group accuracy scatter)
> > * Figure 3: https://i.postimg.cc/bwQvWZQr/Fig3.png (model performance frontiers)
> > * Figure 4: https://i.postimg.cc/vBmH4XCx/Fig4.png (model disparity frontiers)
> > * Figure 5: https://i.postimg.cc/xdnd29RR/Fig5.png (metrics correlation scatter)
> > * Figure 6: https://i.postimg.cc/nL2hVNSt/Fig6.png (model selection effects bar plots)
> > * Figure 7: https://i.postimg.cc/QN9027Mj/Fig7.png (hyperparameter sens. summary)
> > * Figure 8: https://i.postimg.cc/Dy15ZKsD/Fig8.png (hyperparameter sens. detail)

---

> > > ### Comment · Reviewer_kdGJ · 2022-08-08
> > > **Thank you and additional question**
> > >
> > > Thank you very much for the thorough and detailed responses. This clarifies much of my concerns. One area that is still not entirely clear: in the updated revision where you recommend trees as a default--is it necessary to do hyperparameter sweeps? if so, it might be helpful to add a few remarks based on your experiments as to which parameters and how fine-tuned a grid one must use.

---

> > > > ### Author Response · Authors · 2022-08-08
> > > > **Author response to additional question (1/2)**
> > > >
> > > > Thank you for the follow-up comment; we are glad that the response addressed many of your concerns.
> > > >
> > > > The question the reviewer raises about hyperparameter sweeps is a good one that our results are also well-positioned to address. We note that whether hyperparameter sweeps are “necessary” in any given application depends on a variety of factors including compute resources, costs of lowered accuracy/increased disparity due to suboptimal hyperparameter settings, and training/sweep time. So while there is no definitive answer that applies to all settings (even those covered in our work), we can provide some general insights drawn from our results.
> > > >
> > > > ## A. Are hyperparameter sweeps necessary?
> > > >
> > > > With respect to the reviewers’ question, our results suggest the following:
> > > >
> > > > ### 1. Model performance still varies over hyperparameter configurations, even for tree-based models.
> > > >
> > > > For example, Figure 1a shows the variability across different hyperparameter settings in XGBoost models on ACS Income task, and Figure 2 shows considerable variability over the various hyperparameter/algorithm points in the “tree-based models” category (orange points in Figure 2). Supplementary Figures 10, 11, and 12 also show the variability over hyperparameter settings, by model, for each dataset in our study. These results partially reflect the size of the (very large) hyperparameter grids used in our study, as we attempt to cover or exceed the hyperparameter grids swept in each baseline models’ original study/paper.
> > > >
> > > > The Pareto curves traced in Figure 1(c, d) and Figures 3,4 give an intuitive measure of the sensitivity of the models on the frontier to hyperparameters that is also less sensitive to the overall size of the grid we sweep over. We can interpret the range of the curve (distance covered by its points along either x- or y-axes) in either plot as a measure of the performance improvements from hyperparameter tuning for the “best” models within each class (i.e. those along its performance frontier). If each curve was very narrow (had small range along both axes), there would be limited benefit to tuning a model; however, this is not the case in our experiments.
> > > >
> > > > ### 2. Tree-based models are far less sensitive to hyperparameter settings than the other models evaluated.
> > > >
> > > > Our new results, shown in Figures 7, 8, provide strong statistical evidence to support this claim. These figures show the performance of all models in the (truncated) hyperparameter sweep grid, in ascending order, relative to the best-performing model (i.e. the Group DRO model with best worst-group accuracy corresponds to the rightmost point on the Group DRO line), with 95% Clopper-Pearson CIs. For example, in Figure 7, the CIs for XGBoost lie completely outside and above those for Group DRO and DRO $\chi^2$ on four datasets (ACS Income, Adult, LARC, ACS Public Coverage). XGBoost also outperforms these models on Communities and Crime, Compas, and German Credit, although the CIs are much wider due to the small test set sizes in these datasets. We note that these results in Figures 7,8 are after removing hyperparameter values which performed poorly across all datasets for each model, so they account for the wide range of hyperparameter sweeps by retaining only regions with good performance on at least one dataset. Our method for conducting the model sensitivity analysis is described in Section F.2; We will add more detail on this truncation process and include the method and results for this analysis in the main text with the additional page allowed in the updated version of the paper.
> > > >
> > > > We note that as a practical consequence of these results, tree-based models with a randomly-chosen set of hyperparameters from our grid are likely to perform closer to their best-possible performance than non-tree models (e.g. $\chi^2$ DRO, Group DRO) with a randomly-chosen set of hyperparameters from our grid -- that is, even in the absence of tuning, tree-based models are likely to perform better than non-tree models, relative to their maximal empirical performance.

---

> > > > > ### Author Response · Authors · 2022-08-08
> > > > > **Author response to reviewer question (2/2)**
> > > > >
> > > > >
> > > > > ## B. Guidance on sweep parameters and grids
> > > > >
> > > > > We appreciate the reviewers’ suggestion that we “add a few remarks based on your experiments as to which parameters and how fine-tuned a grid one must use”. We think that this is an excellent suggestion! We agree that it would be useful to provide practical guidance on how to construct (or grow/prune) hyperparameter grids, and which parameters were most important to tune. We also believe that providing a recommended set of default hyperparameters for each model would be a useful contribution our large-scale study is poised to make.
> > > > >
> > > > > 1. We will add more detailed discussions of these considerations in our updated version. We will also give the complete hyperparameter configurations for the “best” models, and their exact performance, according to (i) overall accuracy, (ii) worst-group accuracy, and (iii) CVaR risk. Additionally, we note that the complete grid we used for all experiments is currently available in Table 2 in the supplement.
> > > > >
> > > > > 2. Another insight our study stands to provide is an assessment of the existing default settings of models (where such defaults exist), and a recommended set of defaults, based on hyperparmaeters which performed well *across* datasets in our study. To that end, we will also provide (i) a clear reporting of the exact performance of the default hyperparameters for each model used in our study, and (ii) a suggested set of defaults for each model, based on best average performance across datasets in our study, for multiple performance metrics.
> > > > >
> > > > >
> > > > > Direct links to updated figures from main text:
> > > > >
> > > > > * Figure 1: https://i.postimg.cc/fR4L8cS1/Fig1.png (main results summary)
> > > > > * Figure 2: https://i.postimg.cc/NMPGmmwJ/Fig2.png (overall vs. worst-group accuracy scatter)
> > > > > * Figure 3: https://i.postimg.cc/bwQvWZQr/Fig3.png (model performance frontiers)
> > > > > * Figure 4: https://i.postimg.cc/vBmH4XCx/Fig4.png (model disparity frontiers)
> > > > > * Figure 7: https://i.postimg.cc/QN9027Mj/Fig7.png (hyperparameter sens. summary)
> > > > > * Figure 8: https://i.postimg.cc/Dy15ZKsD/Fig8.png (hyperparameter sens. detail)
> > > > > * Figure 10: https://i.postimg.cc/k4LtB92Q/Fig10.jpg (detailed results scatter by alg. 1/3)
> > > > > * Figure 11: https://i.postimg.cc/zDRg4LQ0/Fig11.jpg  (detailed results scatter by alg. 2/3)
> > > > > * Figure 12: https://i.postimg.cc/MHRfB4Bh/Fig12.jpg  (detailed results scatter by alg. 3/3)

---

### Author Response · Authors · 2022-08-02
**Overall Response to All Reviewers, Part 1**

We thank all reviewers for their thoughtful and receptive comments on our work. We are encouraged that they found our work original (kdGJ, WCHM) and needed by the community (kdGJ, WCHM) with the potential for positive impact on the field (kdGJ, Uv7V).

Below, we provide a response to the key shared themes raised across the reviews, as we believe a single response best addresses some common concerns. Additionally, we provide individual responses to the specific concerns raised in each review in individual threaded replies below each review.

The following themes were raised by multiple reviewers:

# 1. Explanation or intuition for *why* tree-based models show better subgroup robustness.

We agree that this is a potentially useful extension to this work. Our primary focus is on a broad evaluation to rigorously compare the baseline robustness of SOTA robust learning methods to effective tabular baselines and identify relevant phenomena. Targeted experiments to precisely verify potential causes for the improved subgroup robustness of tree-based models is beyond the scope of the current work. We believe that there is one clear overall explanation, which all reviewers also identified. We offer two potential hypotheses for future work:

* *Overall Intuition: Trees fit the data better.* In this case, the subgroup robustness is a consequence of the more general and well-known trend that tree-based models show comparable or better performance on tabular data, widely documented in the works cited in our paper, i.e. [1, 2, 3, 4]. Our findings can be also viewed as an analogue of the empirical relationship between in-distribution accuracy and out-of-distribution accuracy in computer vision  documented in [5]. However, [5] only experiments with  image (not tabular) datasets, so our results extend the phenomenon identified by [5] to a new domain; none of [1, 2, 3, 4] evaluates subgroup performance or robustness.

* *Hypothesis 1: The inductive bias of trees is more suited to complex interactions with sensitive features in tabular data.* Intuitively, a tree-based model has a different bias than e.g. an MLP. It’s possible to think of a tree as having very “deep” (multi-layered) interactions between a specific subset of features (those selected for splitting at each node), or for certain inputs (those with high boosting weights), compared to an MLP. The discrete nature of many sensitive subgroup variables in tabular data makes this of particular relevance for tree models, which can easily split on such variables. It is possible that this functional bias enables trees to model the different interaction effects of tabular features for real-world sensitive subgroups more effectively.

* *Hypothesis 2: Ensembling may be a factor.* Another important property of the tree-based models (XGBoost, CatBoost, Random Forest) in our study is that these are ensemble models. To our knowledge, there are no previous evaluations of the subgroup robustness properties of ensembles, but it is possible that ensembling contributes to improved subgroup robustness. However, we think this hypothesis is less plausible: we are not aware of empirical evidence of ensembling leading to distributional robustness in prior work (particularly when the base learners are trained on the same dataset or distribution).

We have added a discussion of the above points to the main text, in Section 7.

# 2. Recommendations for Practice

Reviewers kdGJ and WCHM requested more concrete recommendations or takeaways for practice. We agree that our study provides important evidence for practitioners.

* *Use tree-based models for tabular classification by default.* Our results clearly suggest that tree-based models are likely to achieve better accuracy and better subgroup robustness than other existing methods, with less hyperparameter tuning and compute, than robust models, for a variety of tabular data tasks. We suggest using tree-based models when no other considerations necessitate the use of distributionally robust models with formal guarantees.
* *Explore non-MLP architectures.* It seems unlikely that simply scaling MLPs (e.g., to more layers) will improve their robustness when using DRO-based methods. Instead, we suggest future research into robust learning with models that have similar structure/bias to the tree-based tabular models. We concur with Reviewer Uv7V that this is a natural direction for future work and a limitation of prior work; we now state this explicitly in our main text. We believe that this is an example of the type of research direction that large-scale empirical evaluations like ours can support. See also our comments in (3) below regarding the MLP-only dimension of this work.

We included new text in the paper (abstract and in Section 7) to foreground these recommendations.

---

> ### Author Response · Authors · 2022-08-02
> **Overall Response to All Reviewers, Part 2**
>
> # 3. Limitations of this study
>
> All reviewers highlighted a need for a more nuanced discussion of our study’s limitations. We have added this to Section 7, but also provide a more detailed discussion here.
>
> * *Statistical uncertainty:* Our study makes an empirical claim based on a large set of observations of model performance. As Reviewer kdGJ notes, our initial presentation of results did not fully quantify the statistical uncertainty inherent to our analysis. We have now updated all figures in the main text such that all summary analyses include proper confidence intervals (Clopper-Pearson or difference-in-proportions, as appropriate, with $\alpha = 0.05$). These improvements strengthen our claims, showing that e.g. XGBoost achieves accuracy-fairness performance frontiers either statistically indistinguishable from, or superior to, existing SOTA robust models (see Figures 1, 3, 4; direct links below).
> * *Hyperparameter search space:* It is always possible that there exist larger search spaces we could have covered for our models that would lead to better performance/robustness, as the space of potential hyperparameters is infinite. However, we cover a *wider* superset of the model/hyperparameter spaces considered in any of the previous robustness works (DRO, Group DRO, DORO, etc.) and also wider than most analyses of tree-based models, so we believe this is unlikely without a qualitative change in models (e.g. *very* deep MLPs), and (as noted in the updated text) we increased our initial hyperparameter grids to ensure that parameters of the best-performing models were not on the edge of continuous grids. We also added Figures 7 and 8 (direct links below) to evaluate the hyperparameter sensitivity of each model class. These new figures also support our conclusions regarding the decreased sensitivity of tree-based models to hyperparameters relative to robustness-enhancing models with strong statistical evidence.
> * *Cannot prove a null hypothesis:* We note that it is not possible to prove (via experiments alone) that robust learning methods (e.g. DRO) cannot learn subgroup-robust classifiers. Instead, we make the more modest claim that, from our comprehensive empirical results, we cannot demonstrate that models trained with “robust” learning techniques have superior subgroup robustness to baseline methods such as tree-based ensembles (equivalently, we cannot reject the hypothesis that tree-based models such as XGBoost have similar subgroup robustness to “robust” learners).
> MLP-only: Our study only evaluates MLP architectures for all NN-based models. We do so because, as Uv7V notes, these are the only functional form used in previous robustness works (DRO, Group DRO, DORO), making them the most reasonable comparison benchmark for robust optimizers. We also note that MLPs have been shown capable of achieving excellent performance on tabular data [6], and that even “improved” neural tabular data models often show limited empirical gains in large-scale evaluations and are still outperformed by (non-NN) tree-based models [1, 2].
>
> ## References:
>
> [1] Ravid Shwartz-Ziv and Amitai Armon. Tabular data: Deep learning is not all you need.444
> Information Fusion, 81:84–90, 2022
>
> [2] Borisov, Vadim et al. Deep neural networks and tabular data: A survey. arXiv preprint arXiv:2110.01889,332 2021.
>
> [3] Huang, Xin, et al. "Tabtransformer: Tabular data modeling using contextual embeddings." arXiv preprint arXiv:2012.06678 (2020).
>
> [4] Gorishniy, Yury, et al. "Revisiting deep learning models for tabular data." Advances in Neural Information Processing Systems 34 (2021): 18932-18943.
>
> [5] Miller, John P., et al. "Accuracy on the line: on the strong correlation between out-of-distribution and in-distribution generalization." International Conference on Machine Learning. PMLR, 2021.
>
> [6] Kadra, Arlind, et al. "Well-tuned simple nets excel on tabular datasets." Advances in neural information processing systems 34 (2021): 23928-23941.
>
> ## Direct links to updated figures from main text:
>
> * Figure 1: https://i.postimg.cc/fR4L8cS1/Fig1.png (main results summary)
> * Figure 2: https://i.postimg.cc/NMPGmmwJ/Fig2.png (overall vs. worst-group accuracy scatter)
> * Figure 3: https://i.postimg.cc/bwQvWZQr/Fig3.png (model performance frontiers)
> * Figure 4: https://i.postimg.cc/vBmH4XCx/Fig4.png (model disparity frontiers)
> * Figure 5: https://i.postimg.cc/xdnd29RR/Fig5.png (metrics correlation scatter)
> * Figure 6: https://i.postimg.cc/nL2hVNSt/Fig6.png (model selection effects bar plots)
> * Figure 7: https://i.postimg.cc/QN9027Mj/Fig7.png (hyperparameter sens. summary)
> * Figure 8: https://i.postimg.cc/Dy15ZKsD/Fig8.png (hyperparameter sens. detail)

---

### Author Response · Authors · 2022-08-08
**Authors checking on additional clarifications or concerns regarding rebuttal**

Hello Reviewers kdGJ, WHCM and Uv7V,

We would like to reach out again to check if there were any additional questions or concerns about our rebuttal that we can address before the reviewer-author discussion period ends on August 9.

Thanks again for taking the time to read our work and provide helpful feedback!

Paper Authors

---

### Meta-Review · Area_Chair_LMjA · 2022-08-30

**Recommendation:** Accept
**Confidence:** Less certain

**Metareview:**

This work looks at sensitive subgroups' propensity to be incorrectly scored across both "traditional" tree-based models and other approaches from the recent literature, on tabular data.  I believe Reviewer kdGJ's numeric score is lower than it should be (thank you, Reviewer kdGJ for a technically strong review and for participating in a back-and-forth with the authors), but I also agree with the concerns that were surfaced here.  I also agree with the questioning of Reviewer WCHM w.r.t. domain generalization, and believe that this is more central to the story of the paper than is presently written.  A borderline paper, I would encourage the authors to seriously consider both of those reviewers' threads in a camera-ready or next submission.

**Award:**

No

---

### Decision · Program_Chairs · 2022-09-14

Accept